# Probabilistic Smoothing with Ratio-Monotone Transforms for Global Optimization

## Abstract

Probabilistic smoothing is a standard tool for global optimization, but existing methods rely on Gaussian kernels and specific transforms, often resulting in strong hyperparameter sensitivity and limited robustness. We propose a general smoothing framework that combines flexible symmetric unimodal kernels with monotonic ratio-based transformations. Under mild conditions, we show that the smoothed objective preserves the global maximizer and that all stationary points concentrate near the true optimum for sufficiently large amplification, without requiring a decreasing smoothing schedule. We further provide explicit complexity bounds for stochastic gradient ascent and show that a leave-one-out baseline provably reduces variance. Experiments on high-dimensional benchmarks and black-box adversarial attacks demonstrate improved robustness and competitive performance.

## 1. Introduction

Global optimization over a compact domain is a long-standing challenge in machine learning and engineering, especially when the objective function is highly nonconvex, multimodal, or accessible only through black-box evaluations (Naser et al., 2025). In such settings, gradient-based methods are sensitive to initialization and often converge to suboptimal local extrema.

A classical approach to global optimization is *homotopy-based optimization*, also known as continuation or graduated non-convexity methods, which replace the original objective with a smoothed surrogate and gradually recover the original landscape. Among these, *Gaussian homotopy*, which smooths the objective via Gaussian convolution, has been widely used in vision, inverse problems, and adversarial robustness (Blake & Zisserman, 1987; Brox & Malik, 2011; Szegedy et al., 2013; Mobahi & Fisher III, 2015; Hazan et al., 2016). Classical Gaussian homotopy relies on an explicit continuation schedule implemented through a multi-loop procedure, leading to substantial computational overhead and sensitivity to hyperparameter tuning. To mitigate this cost, *single-loop Gaussian smoothing* methods update the optimization variable and smoothing scale simultaneously (Iwakiri et al., 2022). While these methods reduce computational burden, their theoretical guarantees remain local, characterizing convergence only to stationary points of the smoothed objective rather than the global optimum.

A major advance within Gaussian smoothing was introduced by Xu (2025), which replaces explicit continuation by applying power or exponential transformations before smoothing. This amplification forces stationary points of the smoothed objective to concentrate near the global maximizer, yielding the first approximate global optimality guarantee in a single-loop Gaussian smoothing setting. However, strong amplification also sharpens curvature and inflates the variance of Monte Carlo gradient estimators, resulting in pronounced sensitivity to the amplification parameter and a clear trade-off between localization strength and optimization stability.

In parallel, zeroth-order optimization has shown that Gaussian smoothing enables gradient-free optimization via Monte Carlo score-function estimators (Ghadimi & Lan, 2013; Nesterov & Spokoiny, 2017; Chen et al., 2019). While subsequent work generalized the perturbation distribution beyond Gaussian kernels to reduce estimator variance (Gao & Sener, 2022), these approaches provide guarantees only for local stationarity and do not address global localization.

This paper revisits probabilistic smoothing from a unified perspective. We view Gaussian homotopy, single-loop smoothing, and amplification-based methods as instances of a broader design space defined by (i) the smoothing distribution and (ii) the transformation applied to objective values prior to smoothing. From this viewpoint, instability arises not from smoothing itself, but from restrictive kernel choices combined with overly aggressive transformations.

Motivated by this observation, we propose *Probabilistic*

[1] Anonymous Institution, Anonymous City, Anonymous Region, Anonymous Country. Correspondence to: Anonymous Author <anon.email@domain.com>.

Preliminary work. Under review by the International Conference on Machine Learning (ICML). Do not distribute.

*Table 1.* Comparison of representative zeroth-order optimization methods. The table summarizes the evaluation setting (stochastic or deterministic), the type of optimality guarantee (local stationarity or approximate global localization), the required assumptions on the objective $f$, and the iteration complexity to achieve $\|\nabla f\|^2 \leq \varepsilon$.

| Method | Sto. / Det. | Optimality | Assumptions on $f$ | Iteration Complexity |
|---|---|---|---|---|
| RSGF (Ghadimi & Lan, 2013) | Stochastic | Local | $f$ smooth, Lipschitz gradient | $\mathcal{O}(d/\varepsilon^2)$ |
| ZO-SGD (Nesterov & Spokoiny, 2017) | Stochastic | Local | $f$ smooth or weakly smooth | $\mathcal{O}^a(M/\varepsilon^2)$ |
| ZO-AdaMM (Chen et al., 2019) | Stochastic | Local | $f$ smooth, Lipschitz gradient | $\mathcal{O}(d^2/\varepsilon^2)$ |
| ZO-SLGH (Iwakiri et al., 2022) | Deterministic | Local | $f$ smooth, Lipschitz gradient | $\mathcal{O}^b(d/\varepsilon)$ |
| ZO-SLGH (Iwakiri et al., 2022) | Stochastic | Local | $f$ smooth, Lipschitz gradient | $\mathcal{O}^b(d/\varepsilon^2)$ |
| BeS (Gao & Sener, 2022) | Stochastic | Local | $f$ smooth, bounded; biased gradients | $\mathcal{O}^a(M/\varepsilon^2)$ |
| EPGS (Xu, 2025) | Deterministic | Approx. global | $f$ bounded on compact domain | $\mathcal{O}(d^4/\varepsilon^2)$ |
| Ours | Deterministic | Approx. global | $f$ bounded on compact domain | $\mathcal{O}(d^2/\varepsilon^2)$ |
| Ours with Variance Reduction | Deterministic | Approx. global | $f$ bounded on compact domain | $\mathcal{O}^c\big((1 - C/2)\,d^2/\varepsilon^2\big)$ |

[a] The dependence on the dimension $d$ is not explicit. Here, $M$ denotes an upper bound on the second moment of the stochastic gradient estimator, which may implicitly depend on $d$.
[b] The original results in Iwakiri et al. (2022) are stated for $\|\nabla f\| \leq \varepsilon$; all bounds are converted to $\|\nabla f\|^2 \leq \varepsilon$ for consistency.
[c] The constant $C$ depends on the objective function $f$ and algorithmic hyperparameters, but is independent of $\varepsilon$. While this does not change the asymptotic order of the iteration complexity, the exact bound is strictly smaller since $(1 - C/2) < 1$.

*Smoothing with Ratio-Monotone Transforms (ProMoT)*, a single-loop framework that generalizes both components. We allow a broad class of symmetric unimodal smoothing kernels, including heavy-tailed distributions, and introduce ratio-monotone transformations that subsume power and exponential forms. Under mild conditions, we show that the global maximizer of the original objective is preserved and that all stationary points of the smoothed objective concentrate near the true optimum without requiring a decreasing smoothing schedule.

Finally, since probabilistic smoothing relies on Monte Carlo gradient estimation, its variance is a fundamental bottleneck. We introduce *ProMoT-loo*, a variance-reduced variant based on a leave-one-out baseline. We show that this estimator is unbiased and yields a strictly smaller second-moment bound, and we explicitly quantify how this variance reduction improves the iteration-complexity. We summarize our contributions as follows:

- We identify explicit conditions on both smoothing distributions and transformations, covering a broad class of symmetric unimodal kernels, including heavy-tailed distributions beyond Gaussian, and introducing ratio-monotone transformations that strictly generalize the power and exponential transforms of Xu (2025).

- Under the proposed distributional and transformation conditions, ProMoT achieves the same $\varepsilon$-approximate global localization guarantees as Xu (2025).

- We first introduce a leave-one-out variance reduction scheme for gradient estimation and demonstrate that it strictly improves the iteration-complexity constant.

- We show that ProMoT is significantly more robust to hyperparameter choices than existing methods, with strong performance on high-dimensional benchmarks.

## 2. Related Work

**Gaussian homotopy methods.** Smoothing-based optimization replaces the original objective with a smoothed surrogate, most commonly via Gaussian convolution. Classical Gaussian homotopy or graduated optimization methods rely on a continuation schedule that gradually decreases the smoothing scale to recover the original landscape (Blake & Zisserman, 1987; Mobahi & Fisher III, 2015; Hazan et al., 2016). While effective in practice, such multi-loop procedures incur additional computational overhead and are sensitive to the choice of the smoothing schedule.

To mitigate this issue, single-loop Gaussian homotopy (SLGH) methods update the optimization variable and the smoothing scale simultaneously (Iwakiri et al., 2022). Under smoothness and Lipschitz-gradient assumptions on $f$, deterministic and stochastic variants of SLGH guarantee convergence to stationary points with iteration complexity $\mathcal{O}(d/\varepsilon)$ and $\mathcal{O}(d/\varepsilon^2)$, respectively. As reported in Table 1, these guarantees are purely local, characterizing convergence to stationary points of the smoothed objective rather than to the global maximizer of $f$.

A further line of work removes explicit continuation schedules by modifying the objective prior to Gaussian smoothing. In particular, exponential and power-based Gaussian smoothing (EPGS) was the first to provide a global optimality guarantee in a Gaussian smoothing framework (Xu, 2025), with iteration complexity $\mathcal{O}(d^4/\varepsilon^2)$.

**Non-Gaussian kernels and zeroth-order smoothing.** Zeroth-order optimization enables gradient-free learning by estimating gradients through randomized perturbations. Early work established convergence to stationary points under Gaussian smoothing for nonconvex objectives (Ghadimi & Lan, 2013; Nesterov & Spokoiny, 2017), with iteration

complexity on the order of $\mathcal{O}(d/\varepsilon^2)$. Subsequent studies generalized the perturbation distribution beyond Gaussians to reduce estimator variance and improve sample efficiency (Chen et al., 2019; Gao & Sener, 2022). These approaches retain local stationarity guarantees and typically require smoothness or weak smoothness assumptions on $f$.

**Relation to the present work.**  The proposed approach builds on amplification-based probabilistic smoothing and extends prior work in two key directions. First, we generalize both the smoothing kernel and the transformation while preserving approximate global localization on compact domains. Second, we derive explicit iteration-complexity bounds that quantify the impact of variance reduction in generalized smoothing, showing that it improves the complexity constant without changing the asymptotic rate, as summarized in Table 1.

# 3. Probabilistic Smoothing with Ratio-Monotone Transforms

Consider a compact set $\mathcal{S} \subset \mathbb{R}^d$ and a continuous function $f : \mathcal{S} \to \mathbb{R}$. The global optimization problem is

$$\max_{\mathbf{x} \in \mathcal{S}} f(\mathbf{x}), \tag{1}$$

and $\mathbf{x}^*$ is a global maximizer (i.e., $\mathbf{x}^* \in \arg\max_{\mathbf{x} \in \mathcal{S}} f(\mathbf{x})$). Direct gradient ascent on $f$ can converge to a local maximizer depending on the initialization, which motivates smoothing-based approaches.

Gaussian homotopy (Mobahi & Fisher III, 2015) optimizes a smoothed surrogate

$$G_\sigma(\boldsymbol{\mu}) := \mathbb{E}_{\mathbf{X} \sim \mathcal{N}(\boldsymbol{\mu}, \sigma I)}\big[f(\mathbf{X}); \mathcal{S}\big], \tag{2}$$

and gradually decreases the smoothing scale $\sigma > 0$ so that maximizers of $G_\sigma$ track the global maximizer $\mathbf{x}^*$. While effective, such continuation schemes require carefully tuned multi-loop schedules and are computationally sensitive (Iwakiri et al., 2022). To remove explicit continuation, Xu (2025) proposed amplifying the objective prior to smoothing, enabling single-loop optimization with fixed $\sigma$. Specifically, power and exponential transforms,

$$G_{\theta,\sigma}^{\text{PGS}}(\boldsymbol{\mu}) = \mathbb{E}[f(\mathbf{X})^\theta; \mathcal{S}], \quad G_{\theta,\sigma}^{\text{EPGS}}(\boldsymbol{\mu}) = \mathbb{E}[e^{\theta f(\mathbf{X})}; \mathcal{S}], \tag{3}$$

concentrate stationary points near $\mathbf{x}^*$ for large $\theta$. However, stronger amplification increases curvature and gradient variance, leading to a stability–localization trade-off and increased sensitivity to parameter tuning, a phenomenon that will be explicitly illustrated through a motivating example in a later section.

The present formulation generalizes both lines along two axes. First, the Gaussian kernel is replaced by a general product density that can include even long-tail distribution. Let $p$ be a one–dimensional probability density on $\mathbb{R}$, define

the product kernel $\mathbf{p}(\mathbf{z}) := \prod_{i=1}^d p(z_i)$, and let

$$\mathbf{p}_{\boldsymbol{\mu},\sigma}(\mathbf{x}) := \sigma^{-d} \prod_{i=1}^d p\big((x_i - \mu_i)/\sigma\big) \tag{4}$$

denote the same kernel *shifted to center $\boldsymbol{\mu}$ and rescaled by $\sigma$* (i.e., the original kernel moved to $\boldsymbol{\mu}$ with bandwidth $\sigma$).

Second, the power/exponential transform is replaced by a general transform class that preserves order and expands level ratios. For an amplification parameter $\theta > 0$ and a strictly increasing transform $g(\theta, \cdot)$, the transformed probabilistic smoothing is defined as

$$G_{\theta,\sigma}(\boldsymbol{\mu}) := \mathbb{E}_{\mathbf{X} \sim \mathbf{p}_{\boldsymbol{\mu},\sigma}}\big[g\big(\theta, f(\mathbf{X})\big); \mathcal{S}\big]. \tag{5}$$

Choosing $p$ as the standard Gaussian and $g(\theta, y) = y$ recovers Gaussian homotopy (Mobahi & Fisher III, 2015); choosing $p$ as Gaussian and $g(\theta, y) = y^\theta$ or $e^{\theta y}$ recovers power or exponential variants (Xu, 2025); the formulation in (5) subsumes both by allowing general $p$ and a general ratio-monotone transform $g$. Our goal is to generalize the kernel $p$ and transform $g(\theta, \cdot)$ so that maximizing $G_{\theta,\sigma}(\boldsymbol{\mu})$ is equivalent to maximizing $f(\mathbf{x})$ and admits a unique global maximizer reachable by gradient ascent.

**Optimization of the Smoothed Objective**  We next describe how the generalized smoothed objective $G_{\theta,\sigma}$ can be optimized in practice. Under mild regularity conditions on the smoothing kernel, the gradient of $G_{\theta,\sigma}$ with respect to the location parameter $\boldsymbol{\mu}$ admits a score-function representation,

$$\nabla_{\boldsymbol{\mu}} G_{\theta,\sigma}(\boldsymbol{\mu}) = \mathbb{E}_{\mathbf{X} \sim \mathbf{p}_{\boldsymbol{\mu},\sigma}}\big[g\big(\theta, f(\mathbf{X})\big) \nabla_{\boldsymbol{\mu}} \log \mathbf{p}_{\boldsymbol{\mu},\sigma}(\mathbf{X}) ; \mathcal{S}\big], \tag{6}$$

which enables gradient estimation without requiring derivatives of $f$. Throughout this work, we adopt this formulation to derive stochastic optimization algorithms for $G_{\theta,\sigma}$.

Given a mini-batch $\{\mathbf{X}^{(k)}\}_{k=1}^B \sim \mathbf{p}_{\boldsymbol{\mu},\sigma}$, define $S(\mathbf{x}) = \nabla_{\boldsymbol{\mu}} \log \mathbf{p}_{\boldsymbol{\mu},\sigma}(\mathbf{x})$ and $h(\mathbf{x}) = g(\theta, f(\mathbf{x})) \mathbf{1}_{\{\mathbf{x} \in \mathcal{S}\}}$. A Monte Carlo estimator of the gradient is then given by

$$\widehat{\nabla}_{\boldsymbol{\mu}} G_{\theta,\sigma}(\boldsymbol{\mu}) = \frac{1}{B} \sum_{k=1}^B h(\mathbf{X}^{(k)}) S(\mathbf{X}^{(k)}). \tag{7}$$

The parameter $\boldsymbol{\mu}$ is updated by stochastic gradient ascent,

$$\boldsymbol{\mu}_{t+1} = \boldsymbol{\mu}_t + \eta_t \widehat{\nabla}_{\boldsymbol{\mu}} G_{\theta,\sigma}(\boldsymbol{\mu}_t), \tag{8}$$

where $\eta_t > 0$ is a step size. Notably, the global localization guarantees established in this work do not rely on decreasing the smoothing scale $\sigma$ over time.

**Coordinate-wise smoothing.**  The formulation readily extends to anisotropic smoothing. Instead of a single isotropic scale $\sigma$, we may use a diagonal matrix $\Sigma = \text{diag}(\sigma_1^2, \ldots, \sigma_d^2)$ and define $\mathbf{p}_{\boldsymbol{\mu},\Sigma}(\mathbf{x}) = \prod_{i=1}^d \sigma_i^{-1} p\big((x_i - \mu_i)/\sigma_i\big)$. This allows different smoothing resolutions across coordinates and is useful when the objective exhibits heterogeneous curvature or sensitivity along different dimensions.

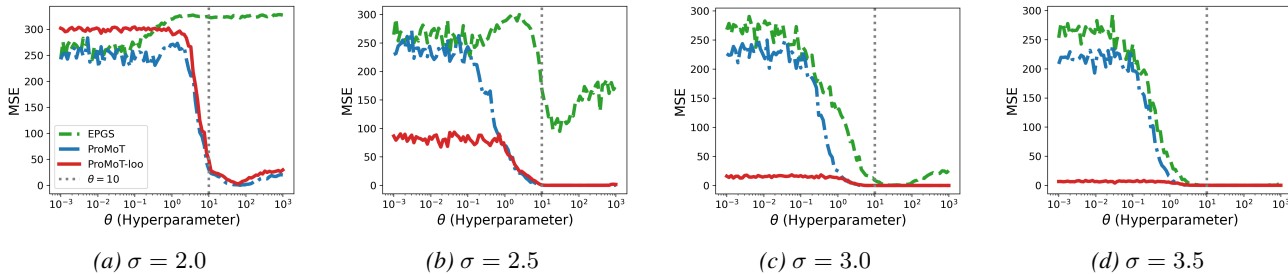

*(a)* $\sigma = 2.0$      *(b)* $\sigma = 2.5$      *(c)* $\sigma = 3.0$      *(d)* $\sigma = 3.5$

*Figure 1.* Mean squared error (MSE) between the true global maximizer $x^*$ of the original objective $f(x)$ and the solution obtained by maximizing the corresponding smoothed objective, plotted as a function of the amplification parameter $\theta$ (log scale). For each value of $\theta$, we run stochastic gradient ascent on the smoothed objective starting from a fixed initialization and record the final estimate $\hat{x}(\theta)$; the reported MSE is $\|\hat{x}(\theta) - x^*\|^2$, averaged over 10 runs.

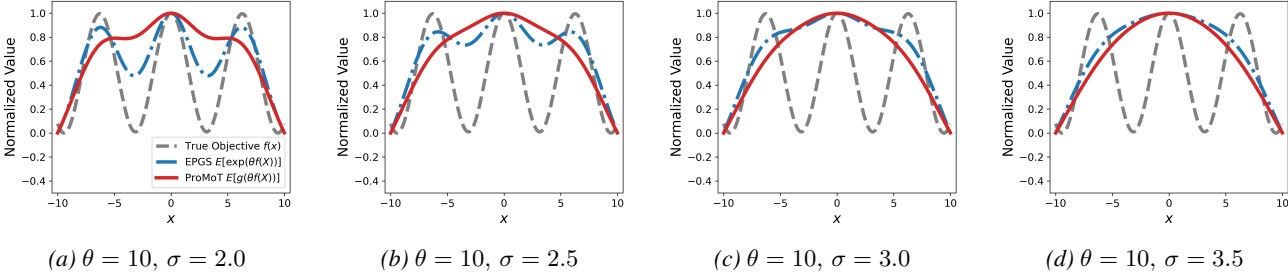

*(a)* $\theta = 10$, $\sigma = 2.0$    *(b)* $\theta = 10$, $\sigma = 2.5$    *(c)* $\theta = 10$, $\sigma = 3.0$    *(d)* $\theta = 10$, $\sigma = 3.5$

*Figure 2.* Effect of probabilistic smoothing on a multimodal one-dimensional objective. The dashed gray curve denotes the original objective $f(x)$. Each colored curve represents a smoothed objective obtained by taking expectations under a shifted smoothing distribution with a fixed amplification parameter $\theta = 10$ and varying the smoothing scale $\sigma$. The value $\theta = 10$ corresponds to the gray dotted vertical reference line in Figure 1, and this figure visualizes how changing $\sigma$ at that fixed $\theta$ reshapes the optimization landscape.

**Variance reduction via leave-one-out baselines.** While amplification improves localization, it can increase the variance of the stochastic gradient estimator. To mitigate this effect, we introduce a leave-one-out baseline that centers each sample contribution using information from the remaining samples in the same mini-batch, while preserving unbiasedness.

For the baseline, define $U := \sum_{j=1}^{B} h(\mathbf{X}^{(j)}) \|S(\mathbf{X}^{(j)})\|^2$ and $V := \sum_{j=1}^{B} \|S(\mathbf{X}^{(j)})\|^2$. For each index $k$, the leave-one-out baseline is given by

$$b_k = \left(U - h(\mathbf{X}^{(k)}) \|S(\mathbf{X}^{(k)})\|^2\right) \Big/ \left(V - \|S(\mathbf{X}^{(k)})\|^2 + \lambda\right), \quad (9)$$

where $\lambda > 0$ is a small ridge parameter for numerical stability. The resulting variance-reduced estimator is

$$\widehat{\nabla}_{\boldsymbol{\mu}}^{\text{loo}} G_{\theta,\sigma}(\boldsymbol{\mu}) = \frac{1}{B} \sum_{k=1}^{B} \left(h(\mathbf{X}^{(k)}) - b_k\right) S(\mathbf{X}^{(k)}). \quad (10)$$

This estimator remains unbiased and admits a strictly smaller second-moment bound than the baseline-free estimator. As shown in the theoretical analysis, this variance reduction leads to an explicit improvement in the iteration complexity constant, without altering the asymptotic convergence rate.

### 3.1. A Motivating Example: Parameter Sensitivity in Probabilistic Smoothing

We illustrate the effect of transformation and kernel generalization using a one-dimensional nonconvex objective $f(\mathbf{x})$ with a unique global maximizer $\mathbf{x}^*$. For each method and each value of the amplification parameter $\theta$, we construct the corresponding probabilistically smoothed surrogate objective by applying the specified transformation and smoothing operator. Starting from a fixed initialization, we run stochastic gradient ascent on each smoothed objective for a fixed number of iterations and record the final estimate $\hat{\mathbf{x}}(\theta)$. The reported error in Figure 1 is the mean squared error $\|\hat{\mathbf{x}}(\theta) - \mathbf{x}^*\|^2$, averaged over multiple independent runs to account for stochasticity.

The key advantage of generalizing both the transformation and the smoothing kernel is improved optimization stability. As shown in Figure 1, exponential-based smoothing (EPGS) achieves low error only within a narrowly tuned range of the amplification parameter $\theta$, and its performance degrades rapidly under mild misspecification. In contrast, ProMoT and ProMoT-loo maintain consistently low error over several orders of magnitude of $\theta$, indicating substantially reduced sensitivity to hyperparameter choices. In particular, ProMoT-loo exhibits pronounced robustness with respect to the choice of $\theta$.

To further understand the origin of this stability, Figure 2 visualizes the smoothed optimization landscapes produced

by different methods. Here, we directly plot the smoothed surrogate objectives obtained from the same base function $f(x)$ at a fixed amplification level $\theta = 10$ while varying the smoothing scale $\sigma$, without performing any optimization. The dashed gray curve denotes the original objective, and each colored curve corresponds to a different choice of $\sigma$.

The figure reveals a qualitative difference in the resulting smoothed landscapes. Under exponential amplification, multiple local extrema persist after smoothing, and increasing $\theta$ amplifies irregularities and leads to unstable gradient behavior. By contrast, combining order-preserving, moderated transformations with more flexible smoothing kernels suppresses spurious extrema and yields an effectively unimodal surrogate. This enhanced landscape regularity explains why ProMoT and ProMoT-loo converge reliably across a wide range of hyperparameters, without delicate tuning.

# 4. Complexity Analysis

We establish $\delta$-approximate global optimality and iteration complexity guarantees under a set of structural assumptions on the smoothing kernel and the transformation. These assumptions are introduced solely for analysis and enable control of smoothness, variance, and global optimality properties of the smoothed objective.

**Assumptions on Smoothing Probability**  To extend the Gaussian smoothing framework to a more general class of smoothing distributions, we identify two fundamental distributional conditions.

**Assumption 4.1** (Fisher Information and Regularity).  Assume $p \in C^2$. Then, there exist two constants, $I := \int_{\mathbb{R}} (p'(z))^2/p(z)\, dz < \infty$ and $K := \int_{\mathbb{R}} |p''(z)|\, dz < \infty$ where $I$ is the Fisher information of $p$ and $K$ reflects a regularity condition controlling the total variation of $p'$.

**Assumption 4.2** (Symmetry and Unimodality).  Assume $p$ is symmetric and unimodal so that $p(z) = p(-z)$ and $p'(z) \le 0$ for $z > 0$ to ensure $\lim_{z \to \infty} p(z) = 0$.

**Theorem 4.1** (Uniform non-degeneracy on a compact window).  *Fix* $\delta > 0$. *For every* $r > 0$ *and every window center* $c$ *such that* $|c| > r + \delta$ *(i.e.,* $[c - r - \delta, c + r + \delta]$ *does not intersect* 0*), there exist constants* $C_\delta(c, r) > 0$ *and* $c_\delta(c, r) > 0$ *such that*

$$\inf_{|x-c| \le r} \left[ F(x + \delta) - F(x - \delta) \right] \ge C_\delta(c, r), \quad (11)$$

$$\inf_{|x-c| \le r} \left| p(x + \delta) - p(x - \delta) \right| \ge c_\delta(c, r). \quad (12)$$

The proof can be found in Appendix B.1. This theorem provides the sign structure and tail behavior needed for localization arguments, ensuring that derivative signs are consistent and that mass vanishes at infinity. Representative smoothing kernels satisfying Assumptions 4.1 and 4.2 are

*Table 2.* Examples of smoothing distributions satisfying Assumptions 4.1 and 4.2.

| Kernel $p(z)$ | $I = \int \frac{(p')^2}{p}$ | $K = \int |p''|$ |
|---|---|---|
| Gaussian $\frac{1}{\sqrt{2\pi}} e^{-z^2/2}$ | 1 | 0.96749 |
| Logistic $e^{-z}/(1 + e^{-z})^2$ | $1/3$ | 0.38496 |
| Student-$t_\nu$ | $\frac{\nu + 1}{\nu + 3}$ | (numeric) |
| $\nu = 1$ (Cauchy) | 0.50000 | 0.82691 |
| $\nu = 3$ | 0.66667 | 0.87870 |
| $\nu = 10$ | 0.84615 | 0.92883 |
| Hyper. Secant $\frac{1}{2}\text{sech}(\pi z/2)$ | $\pi^2/8$ | $\pi/2$ |
| Gen. Gaussian $\beta e^{-|z|^\beta}/(2\Gamma(1/\beta))$ | 4.05587 | 3.36400 |

*Table 3.* Examples of ratio-monotone transformations $g(\theta, y)$.

| $g(\theta, y)$ | $\frac{d}{d\theta}\left( \frac{g(\theta,a)}{g(\theta,b)} \right)$ for $a > b$ |
|---|---|
| $(y + c)^\theta$ | $\frac{(a+c)^\theta}{(b+c)^\theta} \log \frac{a+c}{b+c} > 0$ |
| $e^{\theta y}$ | $(a - b)\, e^{\theta(a-b)} > 0$ |
| $e^{\theta y^\alpha}$ | $(a^\alpha - b^\alpha)\, e^{\theta(a^\alpha - b^\alpha)} > 0$ |
| $(y + c)^\beta e^{\theta y}$ | $(a - b) \frac{(a+c)^\beta}{(b+c)^\beta} e^{\theta(a-b)} > 0$ |
| $\log(1 + e^{\theta y})$ | $\frac{a\, e^{\theta a}(1 + e^{\theta b}) - b\, e^{\theta b}(1 + e^{\theta a})}{(1 + e^{\theta a})(1 + e^{\theta b})} > 0$ |
| $\sinh(\theta y)$ | $\frac{\sinh(\theta a)}{\sinh(\theta b)}\left( a \coth(\theta a) - b \coth(\theta b) \right) > 0$ |

summarized in Table 2, together with the corresponding constants $I$ and $K$ that govern variance and curvature bounds in the subsequent analysis.

**Assumptions on Transformation**  Our framework also extends to a broad class of transformations. Specifically, we impose two structural assumptions on the transformation.

**Assumption 4.3** (Monotonicity).  For any fixed $\theta > 0$, $y \mapsto g(\theta, y)$ is strictly increasing with respect to $y \in \mathbb{R}$.

**Assumption 4.4** (Ratio-Monotonicity).  For any fixed $a > b > 0$, the map $\theta \mapsto g(\theta, a)/g(\theta, b)$ is strictly increasing.

Concrete examples of transformations satisfying Assumption 4.4 are listed in Table 3, illustrating that the condition accommodates classical power and exponential forms as well as more general constructions.

## 4.1. Invariance of Global-Optimum

**Theorem 4.2.**  *Suppose the smoothing kernel* $p(x)$ *and transformation* $g(\theta, y)$ *satisfy Assumption 4.1–4.4. Then, for any* $M > 0$ *and* $\delta > 0$ *such that* $\text{cube}(\mathbf{x}^*; \delta) := \{\mathbf{x}|\forall i \in [d], |x_i - x_i^*| \le \delta\}$ *and* $\text{cube}(\mathbf{x}^*; \delta) \subset \mathcal{S}$, *there exists* $\theta_{\delta,\sigma,M} > 0$, *such that whenever* $\theta > \theta_{\delta,\sigma,M}$, *for any* $\|\boldsymbol{\mu}\|_\infty < M$ *and any* $i \in [d]$, *we have* $\partial G_{\theta,\sigma}(\boldsymbol{\mu})/\partial \mu_i > 0$ *if* $\mu_i < x_i^* - \delta$, *and* $\partial G_{\theta,\sigma}(\boldsymbol{\mu})/\partial \mu_i < 0$ *if* $\mu_i > x_i^* + \delta$. *Here,* $\mu_i$ *and* $x_i^*$ *denote the* $i$*th dimension of* $\boldsymbol{\mu}$ *and* $\mathbf{x}^*$.

*Remark.* See Appendix B.2 for the proof. The inequalities in Theorem 4.2 imply that any stationary point of $G_{\theta,\sigma}$ within the region $\{\|\boldsymbol{\mu}\|_\infty < M\}$ must lie inside the $\text{cube}(\mathbf{x}^*; \delta)$. In other words, the gradient cannot vanish outside this cube,

since each coordinate derivative is strictly positive to the left of $x_i^* - \delta$ and strictly negative to the right of $x_i^* + \delta$. Thus, all stationary points (if any exist) are confined to a neighborhood of the true maximizer $\mathbf{x}^*$. While Xu (2025) analyzes a similar property under an $\ell_2$-ball with Gaussian kernels, such analysis becomes intractable for general product kernels. We instead use an $\ell_\infty$-cube, which better suits the product kernel and coordinate-wise control.

**Theorem 4.3.** *Fix $\sigma > 0$ and $\theta > 0$. Under Assumption 4.1–4.4, if there exists a unique global maximizer, then the objective $G_{\theta,\sigma}(\boldsymbol{\mu})$ admits the global maximizer.*

The proof can be found in Appendix B.3. Theorem 4.3 guarantees that the smoothed objective $G_{\theta,\sigma}$ actually attains its maximum. This existence result does not depend on Gaussian convolution structure and therefore applies to the broader kernel class considered in this work.

### 4.2. Iteration Complexity of ProMoT

This subsection derives an iteration-complexity guarantee for ProMoT by reducing the analysis to two reusable ingredients: a global smoothness constant for $G_{\theta,\sigma}$ and a uniform second-moment bound for the score estimator. We first show that the Lipschitz constant of $\nabla_{\boldsymbol{\mu}} G_{\theta,\sigma}$ is controlled by $K$ (Lemma 4.4). Next, we bound the second moment of the score estimator via $I$ (Lemma 4.5). We apply a standard one-step smoothness inequality and telescope to control the cumulative squared gradient norm (Theorem 4.6), which yields explicit rates under polynomial steps (Corollary 4.7) and their anisotropic variants (Corollaries 4.8). Finally, we incorporate variance reduction via a leave-one-out baseline, proving unbiasedness and deriving an improved second-moment bound that translates into a strictly better complexity (Lemmas 4.9–4.10, Theorem 4.11, Corollaries 4.12).

The following two lemmas characterize the global Lipschitz continuity of the gradient and the second moment of its Monte–Carlo gradient estimator.

**Lemma 4.4.** *Under Assumption 4.1–4.4, the gradient $\nabla_{\boldsymbol{\mu}} G_{\theta,\sigma}(\mu)$ is globally Lipschitz on $\mathbb{R}^d$ with Lipschitz constant $L_\theta = g_{*,\theta}\, d\, K / \sigma^2$, where $g_{*,\theta} := g(\theta, f(\mathbf{x}^*))$.*

**Lemma 4.5.** *Under Assumption 4.1–4.4, its second moment is bounded as $\mathbb{E}\big[\|\hat{\nabla}_{\boldsymbol{\mu}} G_{\theta,\sigma}(\mu)\|^2\big] \leq Q_\theta := d\, I\, g_{*,\theta}^2 / \sigma^2$.*

The proofs can be found in Appendix B.4 and Appendix B.5. Under these lemmas, we can derive the following inequality.

**Theorem 4.6.** *Under Assumption 4.1–4.4, $\nabla_{\boldsymbol{\mu}} G_{\theta,\sigma}$ is $L_\theta$-Lipschitz and the score estimator has a uniform second-moment bound with $Q_\theta$. Consequently, for any horizon $T \geq 1$ and nonnegative steps $\{\eta_t\}_{t=0}^{T-1}$,*

$$\sum_{t=0}^{T-1} \eta_t\, \mathbb{E}\big[\|\nabla_{\boldsymbol{\mu}} G_{\theta,\sigma}(\boldsymbol{\mu}_t)\|^2\big] \ \leq\ g_{*,\theta} - \mathbb{E}\big[G_{\theta,\sigma}(\boldsymbol{\mu}_0)\big]$$

$$+\ \frac{g_{*,\theta}\, d\, K}{\sigma^2} \cdot \frac{d\, I}{2\sigma^2}\, g_{*,\theta}^2 \sum_{t=0}^{T-1} \eta_t^2. \tag{13}$$

The proof can be found in Appendix B.6. Theorem 4.6 establishes a generic inequality for the score-based stochastic ascent applied to $G_{\theta,\sigma}$. In contrast to Gaussian homotopy analyses (Iwakiri et al., 2022; Xu, 2025), this argument does not rely on Gaussian distribution or gradients of the original objective $f(\mathbf{x})$, but applies directly to a general kernel through the score-function formulation.

**Corollary 4.7.** *Under Assumption 4.1–4.4, fix $\sigma > 0$, $M > 0$, and $\delta > 0$. Then there exists $\theta_{\delta,\sigma,M} > 0$ such that for all $\theta \geq \theta_{\delta,\sigma,M}$, ProMoT converges into a local maximizer of $G_{\theta,\sigma}(\boldsymbol{\mu})$ within $\{\|\boldsymbol{\mu}\|_\infty < M\}$ lies in $\mathrm{cube}(\mathbf{x}^*; \delta)$. Moreover, consider the step-size $\eta_t = \sigma^2/d \cdot (t+1)^{-(\frac{1}{2}+\gamma)}$ for $\gamma \in (0, 1/2)$. Then, for any $\varepsilon > 0$, ProMoT achieves $\mathbb{E}\big[\|\nabla_{\boldsymbol{\mu}} G_{\theta,\sigma}(\boldsymbol{\mu}_T)\|^2\big] < \varepsilon$, whenever $T > \Big(C_\gamma d \cdot (g_{*,\theta} + IK\, g_{*,\theta}^3)/(\sigma^2 \varepsilon)\Big)^{\frac{2}{1-2\gamma}}$ where $C_\gamma$ indicates $(\frac{1}{2} - \gamma)/(2^{\frac{1}{2}-\gamma} - 1)$.*

The proof can be found in Appendix B.7. Corollary 4.7 combines Theorem 4.2 with the iteration bound of Theorem 4.6. As a result, ProMoT is guaranteed not only to approach a stationary point of $G_{\theta,\sigma}$, but also to approach one that lies within a prescribed $\delta$-neighborhood of the true global maximizer $\mathbf{x}^*$. Moreover, as $\gamma \to 0$, the iteration complexity bound approaches the optimal rate shown in Table 1.

**Corollary 4.8.** *Under Assumption 4.1–4.4, assume the single base kernel setting with product density $\mathbf{p}_{\mu,\Sigma}(x) = \prod_{i=1}^d \sigma_i^{-1} p\big((x_i - \mu_i)/\sigma_i\big)$. Define $S_2(\Sigma) := \sum_{i=1}^d \sigma_i^{-2}$. Then, for any fixed $(\theta, \Sigma)$:*

$$L_\theta(\Sigma) \ =\ g_{*,\theta}\, K\, S_2(\Sigma), \quad Q_\theta(\Sigma) \ =\ g_{*,\theta}^2\, I\, S_2(\Sigma). \tag{14}$$

*Moreover, consider the step-size $\eta_t = (t+1)^{-(\frac{1}{2}+\gamma)} \cdot S_2(\Sigma)^{-1}$ for any $\gamma \in (0, \frac{1}{2})$. Then, ProMoT achieves $\mathbb{E}\big[\|\nabla_{\boldsymbol{\mu}} G_{\theta,\sigma}(\boldsymbol{\mu}_T)\|^2\big] < \varepsilon$, whenever $T > \Big(C_\gamma \cdot S_2(\Sigma) \cdot (g_{*,\theta} + IK\, g_{*,\theta}^3)/\varepsilon\Big)^{\frac{2}{1-2\gamma}}$.*

The proof can be found in Appendix B.8. Corollary 4.8 shows that ProMoT allows different smoothing scales to be assigned to different coordinates. Each coordinate $i$ is smoothed with its own scale $\sigma_i$, and both the smoothness constant and the second-moment bound depend on the aggregate quantity $S_2(\Sigma) = \sum_i \sigma_i^{-2}$. Coordinates with smaller $\sigma_i$ contribute more strongly to the curvature and variance terms, making explicit how dimension-wise sensitivity affects the convergence rate. In contrast, standard Gaussian smoothing enforces a single isotropic scale across all coordinates.

### 4.3. Iteration Complexity of ProMoT-loo

We now turn to the theoretical analysis of *ProMoT-loo*, which incorporates a leave-one-out baseline into the score-function estimator.

**Lemma 4.9.** *Under Assumption 4.1–4.4, consider the leave-one-out estimator defined in* (10). *Then, conditional on $\boldsymbol{\mu}_t$,*

$$\mathbb{E}\left[\widehat{\nabla}_{\boldsymbol{\mu}}^{\mathrm{loo}} G_{\theta,\Sigma}(\boldsymbol{\mu}_t) \,\middle|\, \boldsymbol{\mu}_t\right] = \nabla_{\boldsymbol{\mu}} G_{\theta,\Sigma}(\boldsymbol{\mu}_t). \quad (15)$$

See Appendix B.9 for the proof. Lemma 4.9 shows that introducing a leave-one-out baseline does not change the expected update direction. This allows variance reduction to be incorporated without biasing the optimization.

**Lemma 4.10.** *Under Assumption 4.1–4.4, define*

$$R_{\theta,\mu,\Sigma}^2 := \frac{\left(\mathbb{E}[h(\mathbf{X})\|S(\mathbf{X})\|^2 \mid \mu]\right)^2}{\mathbb{E}[h(\mathbf{X})^2\|S(\mathbf{X})\|^2|\mu]\,\mathbb{E}[\|S(\mathbf{X})\|^2|\mu]}. \quad (16)$$

*Then, conditionally on $\boldsymbol{\mu}_t$,*

$$\mathbb{E}\left[\left\|\widehat{\nabla}_{\boldsymbol{\mu}}^{\mathrm{loo}} G_{\theta,\Sigma}(\boldsymbol{\mu}_t)\right\|^2 \middle| \mu_t\right] \leq (1 - R_{\theta,\mu_t,\Sigma}^2)\, g_{*,\theta}^2\, I\, S_2(\Sigma)$$
$$+ C_{\mathrm{loo}}(B,\lambda)\, I\, S_2(\Sigma), \quad (17)$$

*where*

$$C_{\mathrm{loo}}(B,\lambda) := \left(\frac{b^\star\lambda}{\mu_V + \lambda}\right)^2 + \frac{2\,\mathrm{Std}(U)}{\lambda\sqrt{s}} + \frac{2\mu_U\,\mathrm{Std}(V)}{\lambda^2\sqrt{s}}$$
$$+ O\left(\frac{1}{\lambda^2 s}\right) + \frac{2\,\mathrm{Var}(U)}{\lambda^2 s} + \frac{4\mu_U^2\,\mathrm{Var}(V)}{\lambda^4 s} + O\left(\frac{1}{\lambda^4 s^2}\right), \quad (18)$$

*with $s := B - 1$ and $\mu_U = \mathbb{E}[h(\mathbf{X})\|S(\mathbf{X})\|^2|\mu]$, $\mu_V = \mathbb{E}[\|S(\mathbf{X})\|^2|\mu]$, $b^\star = \mu_U/\mu_V$. Here, $O(\cdot)$ ignores constant depending only on $U$ and $V$ (but not on $B$ or $\lambda$).*

See Appendix B.10 for the proof. Lemma 4.10 characterizes how variance reduction through a leave-one-out baseline affects the second moment of the score estimator. The leading term shows that the variance (or the second moment) is reduced proportionally to $(1 - R_{\theta,\mu,\Sigma}^2)$, where $R_{\theta,\mu,\Sigma}^2$ measures the alignment between the score magnitude and the transformed objective. The additional term $C_{\mathrm{loo}}(B,\lambda)$ captures the bias induced by using a regularized finite-sample baseline defined in (9), instead of the variance-optimal one $b^\star$, and is explicitly controlled by the ridge parameter $\lambda$.

**Theorem 4.11.** *With the setting of Lemma 4.9 and 4.10, choose $\lambda = (B-1)^{-1/8}$. Then $C_{\mathrm{loo}}(B,\lambda) = O(B^{-1/4})$ and*

$$\mathbb{E}\left[\left\|\widehat{\nabla}_{\boldsymbol{\mu}}^{\mathrm{loo}} G_{\theta,\Sigma}(\boldsymbol{\mu}_t)\right\|^2 \middle| \boldsymbol{\mu}_t\right]$$
$$\leq \left((1 - R_{\theta,\mu_t,\Sigma}^2)\, g_{*,\theta}^2 + O(B^{-1/4})\right) I S_2(\Sigma). \quad (19)$$

*Define $C_{\theta,\Sigma} := \min_{t\in[T]} \mathbb{E}_{\mu_t}\left[R_{\theta,\mu_t,\Sigma}^2\right]$ and $B := \Omega(16/(C_{\theta,\Sigma} g_{*,\theta}^2)^4)$, then,*

$$\mathbb{E}\left[\left\|\widehat{\nabla}_{\boldsymbol{\mu}}^{\mathrm{loo}} G_{\theta,\Sigma}(\boldsymbol{\mu}_t)\right\|^2\right] \leq (1 - C_{\theta,\Sigma}/2)\, g_{*,\theta}^2 I S_2(\Sigma). \quad (20)$$

See Appendix B.11 for the proof. Theorem 4.11 selects an explicit ridge parameter that balances baseline estimation bias and variance. The key implication of Theorem 4.11 is that introducing a leave-one-out baseline strictly reduces the variance of the score estimator compared to the baseline-free estimator. The leading second-moment term is multiplied by $(1 - C_{\theta,\Sigma}/2)$, which is always no larger than one and is strictly smaller whenever the baseline is informative.

**Corollary 4.12.** *Under the same conditions of Theorem 4.11. Let the step size be $\eta_t = S_2(\Sigma)^{-1}(t+1)^{-(\frac{1}{2}+\gamma)}$ with any $\gamma \in (0, \frac{1}{2})$. Then, ProMoT achieves $\mathbb{E}[\|\nabla_{\boldsymbol{\mu}} G_{\theta,\Sigma}(\boldsymbol{\mu}_T)\|^2] < \varepsilon$, whenever*

$$T > \left[\frac{C_\gamma S_2(\Sigma)}{\varepsilon}\left\{g_{*,\theta} + IK\, g_{*,\theta}^3\left(1 - \frac{C_{\theta,\Sigma}}{2}\right)\right\}\right]^{\frac{2}{1-2\gamma}}. \quad (21)$$

The proof can be found in Appendix B.12. Corollary 4.12 shows that the leave-one-out baseline yields a *provable* reduction in the iteration complexity of ProMoT by strictly improving the second-moment bound of the score estimator. Moreover, as $\gamma \to 0$, the iteration complexity bound approaches the rate shown in Table 1. To the best of our knowledge, this is the first analysis that makes the benefit of variance reduction explicit at the level of iteration complexity, rather than variance stabilization alone. This theoretical improvement is consistent with the empirical results in Figure 1, where ProMoT-loo achieves the best convergence behavior among all compared methods.

## 5. Experiments

In this section, we evaluate ProMoT and ProMoT-loo on (i) canonical high-dimensional non-convex optimization benchmarks and (ii) real-world black-box targeted adversarial attack tasks. We compare against the following baselines: EPGS (Xu, 2025), RSGF (Ghadimi & Lan, 2013), ZO-SGD (Nesterov & Spokoiny, 2017), ZO-AdaMM (Chen et al., 2019), ZO-SLGHd/r (Iwakiri et al., 2022), and CMA-ES (Hansen & Ostermeier, 2001; Hansen et al., 2019).

**Evaluation metrics.** For canonical optimization benchmarks, we report: (i) *Mean Squared Error (MSE)*, defined as the minimum squared distance between the optimization trajectory and the true global optimum; (ii) *hitting time*, defined as the iteration at which this minimum MSE is first achieved; and (iii) *best value*, defined as the objective value corresponding to the minimum MSE. All metrics are averaged over independent runs, with standard deviations reported in parentheses.

For adversarial attack experiments, we report: (i) *success rate (SR)*, defined as the fraction of inputs for which a successful attack is achieved; (ii) mean $R^2(\mathbf{x}, \mathbf{x} + \boldsymbol{\mu})$, measuring similarity between the original input and the perturbed

input; and (iii) mean $L_\infty$ norm of the perturbation, capturing the maximum per-coordinate distortion. Higher $R^2$ and lower $L_\infty$ indicate less perceptible adversarial attacks.

*Table 4.* Ackley

| Method | MSE | Hitting Time | Best Value |
|---|---|---|---|
| ProMoT | 0.49(0.04) | 393.60(7.70) | -4.33(0.11) |
| ProMoT-loo | **0.04(0.00)** | 392.60(9.06) | **-1.71(0.05)** |
| EPGS | 5.98(0.93) | 399.50(1.07) | -9.41(0.49) |
| RSGF | 24.90(0.13) | 22.65(38.56) | -13.42(0.69) |
| ZO-SGD | 0.28(0.05) | 397.75(3.30) | -3.37(0.16) |
| ZO-AdaMM | 12.25(0.70) | 395.40(5.65) | -10.76(0.20) |
| ZO-SLGHd | 0.58(0.15) | 398.70(1.62) | -4.32(0.31) |
| ZO-SLGHr | 0.57(0.11) | 399.25(1.22) | -4.33(0.23) |
| CMA-ES | 0.09(0.01) | 397.85(1.35) | -2.61(0.10) |

*Table 5.* Rosenbrock

| Method | MSE | Hitting Time | Best Value |
|---|---|---|---|
| ProMoT | 0.13(0.01) | 398.95(1.50) | -38.63(2.30) |
| ProMoT-loo | **0.02(0.01)** | 396.40(4.66) | **-3.11(0.21)** |
| EPGS | 0.37(0.01) | 399.85(0.36) | -113.43(4.32) |
| RSGF | 3.99(0.01) | 0.85(1.24) | -3622.45(29.14) |
| ZO-SGD | 0.06(0.01) | 400.00(0.00) | -3.81(0.44) |
| ZO-AdaMM | 0.05(0.01) | 400.00(0.00) | -5.95(0.45) |
| ZO-SLGHd | 0.44(0.02) | 400.00(0.00) | -120.81(4.36) |
| ZO-SLGHr | 0.40(0.02) | 400.00(0.00) | -111.46(3.61) |
| CMA-ES | 0.65(0.03) | 96.90(8.04) | -449.42(56.74) |

*Table 6.* Griewank

| Method | MSE | Hitting Time | Best Value |
|---|---|---|---|
| ProMoT | 1.95(0.12) | 399.05(1.02) | -1.03(0.35) |
| ProMoT-loo | **0.23(0.01)** | 394.60(7.53) | 74.71(4.41) |
| EPGS | 3.32(0.17) | 399.30(1.58) | -1.42(0.02) |
| RSGF | 24.86(0.17) | 24.00(29.72) | -4.11(0.02) |
| ZO-SGD | 3.62(0.06) | 400.00(0.00) | -1.45(0.01) |
| ZO-AdaMM | 0.24(0.05) | 397.10(5.10) | **105.38(1.79)** |
| ZO-SLGHd | 2.77(0.16) | 228.00(8.88) | -0.75(0.31) |
| ZO-SLGHr | 2.81(0.17) | 227.35(9.98) | -0.89(0.30) |
| CMA-ES | 1.16(0.11) | 395.70(2.59) | 38.55(4.15) |

### 5.1. Canonical Non-Convex Benchmarks

All canonical benchmark experiments are conducted in a high-dimensional setting with dimension $d = 500$. Across the Ackley, Rosenbrock, and Griewank benchmarks (Tables 4–6), ProMoT-loo consistently achieves the lowest MSE, indicating reliable convergence toward the global optimum across diverse non-convex landscapes. On Ackley and Rosenbrock, it attains both the lowest MSE and the best fitted values, demonstrating robustness to strong multimodality and curvature anisotropy, while on Griewank it maintains the lowest MSE with competitive objective values. One contributing factor to this advantage in high dimensions is the choice of the smoothing kernel. As the dimension increases, Gaussian smoothing concentrates most samples within a narrow neighborhood (e.g., within $3\sigma$), which limits effective exploration, whereas the heavier-tailed kernels used in ProMoT maintain a higher probability of sampling distant regions, enabling more effective global search and improved performance. Overall, these results highlight that combining heavier-tailed smoothing kernels with ratio-monotone

*Table 7.* CIFAR-10

| Method | SR | mean $R^2$ | mean $L_\infty$ |
|---|---|---|---|
| ProMoT | 1 | 0.97(0.02) | 0.28(0.07) |
| ProMoT-loo | 1 | **0.98(0.01)** | **0.20(0.06)** |
| EPGS | 1 | 0.97(0.02) | 0.28(0.07) |
| RSGF | 0.34 | -0.50(1.50) | 1.49(0.54) |
| ZO-SGD | 1 | 0.98(0.02) | 0.25(0.07) |
| ZO-AdaMM | 1 | 0.96(0.03) | 0.31(0.08) |
| ZO-SLGHd | 0.99 | 0.98(0.01) | 0.22(0.06) |
| ZO-SLGHr | 1 | 0.95(0.04) | 0.31(0.09) |
| CMA-ES | 1 | 0.94(0.04) | 0.32(0.04) |

*Table 8.* VitalDB

| Method | SR | mean $R^2$ | mean $L_\infty$ |
|---|---|---|---|
| ProMoT | 1 | 0.97(0.01) | 0.58(0.17) |
| ProMoT-loo | 1 | 0.98(0.01) | **0.57(0.17)** |
| EPGS | 1 | 0.97(0.01) | 0.65(0.18) |
| RSGF | 1 | 0.91(0.04) | 0.92(0.23) |
| ZO-SGD | 1 | 0.83(0.09) | 1.24(0.33) |
| ZO-AdaMM | 1 | 0.96(0.02) | 0.82(0.27) |
| ZO-SLGHd | 1 | 0.87(0.04) | 1.24(0.25) |
| ZO-SLGHr | 1 | 0.95(0.02) | 0.80(0.19) |
| CMA-ES | 1 | **0.99(0.01)** | 0.60(0.27) |

transformations, together with the leave-one-out estimator, yields stable and consistently superior performance.

### 5.2. Black-Box Targeted Adversarial Attacks

We evaluate ProMoT and ProMoT-loo on black-box targeted adversarial attacks against models trained on CIFAR-10 and VitalDB, with results summarized in Tables 7 and 8. For CIFAR-10, the input dimension is $d = 3{,}072$, corresponding to $32 \times 32 \times 3$ images. For VitalDB, the final input dimension after preprocessing is $d = 42$. Across both domains, ProMoT-loo consistently achieves the smallest $L_\infty$ perturbations while maintaining high attack success rates and competitive mean $R^2$, indicating less perceptible adversarial examples. On CIFAR-10, ProMoT-loo outperforms all baselines in terms of $L_\infty$ without sacrificing reconstruction quality, while on VitalDB—a non-smooth tree-based classifier with severe class imbalance—it again yields the smallest perturbations and remains competitive in $R^2$. Notably, although ProMoT-loo attains a slightly lower mean $R^2$ than CMA-ES (by approximately 1%), it achieves about a 5% reduction in mean $L_\infty$, demonstrating improved imperceptibility with comparable reconstruction quality. Overall, these results highlight the robustness of the proposed framework across heterogeneous black-box models and data modalities.

## 6. Conclusion

We introduced ProMoT, a single-loop probabilistic smoothing framework that achieves approximate global optimization using general smoothing distributions and ratio-monotone transformations. We established global localization and iteration-complexity guarantees with provable variance reduction, and validated robustness on high-dimensional benchmarks and black-box adversarial attacks.

## Impact Statement

This paper presents work whose goal is to advance the field of Machine Learning. There are many potential societal consequences of our work, none which we feel must be specifically highlighted here.

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

## A. Verification of Transformation Assumptions

In this section, we verify that the transformation families listed in Table 3 satisfy Assumption 4.3 (measurability, nonnegativity, and monotonicity) and Assumption 4.4 (ratio monotonicity). When applicable, we also explicitly state a boundedness constant $G_\theta$.

Throughout, we fix $\theta > 0$ and assume that $y$ lies in the domain where each transformation is well-defined (e.g., $y > -c$ when $(y + c)$ appears).

**Power:** $g(\theta, y) = (y + c)^\theta$, $c \geq 0$, $y > -c$

**Assumption 4.3.** We have
$$\partial_y g(\theta, y) = \theta(y + c)^{\theta-1} \geq 0 \quad \text{for } \theta > 0,$$
so $g$ is nondecreasing in $y$ and nonnegative. Measurability is immediate.

**Assumption 4.4.** For $a > b > -c$,
$$\frac{g(\theta, a)}{g(\theta, b)} = \left(\frac{a + c}{b + c}\right)^\theta,$$
and since $(a + c)/(b + c) > 1$, the ratio is strictly increasing in $\theta$.

**Exponential:** $g(\theta, y) = e^{\theta y}$

**Assumption 4.3.**
$$\partial_y g(\theta, y) = \theta e^{\theta y} \geq 0,$$
so $g$ is nondecreasing, nonnegative, and measurable.

**Assumption 4.4.** For $a > b$,
$$\frac{g(\theta, a)}{g(\theta, b)} = e^{\theta(a-b)},$$
which is strictly increasing in $\theta$.

**Softplus (log-exp):** $g(\theta, y) = \log(1 + e^{\theta y})$

**Assumption 4.3.** We have
$$\partial_y g(\theta, y) = \theta \, \sigma(\theta y) \geq 0, \qquad \sigma(t) = \frac{1}{1 + e^{-t}},$$
so $g$ is nondecreasing in $y$ for $\theta > 0$. Moreover, $g(\theta, y) \geq 0$ and is measurable, hence Assumption 4.3 is satisfied.

**Assumption 4.4.** For $a > b$, consider
$$R(\theta) := \log g(\theta, a) - \log g(\theta, b) = \log\left(\frac{\log(1 + e^{\theta a})}{\log(1 + e^{\theta b})}\right).$$

It suffices to show that $R'(\theta) \geq 0$. Define $\phi(t) := t \, \sigma(t) - \log(1 + e^t)$. A direct calculation yields
$$R'(\theta) = \frac{\phi(\theta a) - \phi(\theta b)}{\big(\log(1 + e^{\theta a})\big)\big(\log(1 + e^{\theta b})\big)}.$$

Since
$$\phi'(t) = t \, \sigma(t)\big(1 - \sigma(t)\big) \geq 0 \quad \text{for } t \geq 0,$$
the function $\phi$ is increasing on $[0, \infty)$. Thus, for $a > b > 0$ and $\theta > 0$, we have $\phi(\theta a) \geq \phi(\theta b)$ and hence $R'(\theta) \geq 0$. Therefore, $\theta \mapsto g(\theta, a)/g(\theta, b)$ is nondecreasing, establishing Assumption 4.4.

**Remark.** Compared to pure exponential transforms, the softplus form grows exponentially only for large $\theta y$ while remaining nearly linear for small $\theta y$, which moderates amplification and improves gradient conditioning in practice.

**Sigmoid–power (bounded):** $g(\theta, y) = \sigma(\alpha y)^\theta$, $\alpha > 0$

**Assumption 4.3.** Since $\sigma(\alpha y)$ is increasing in $y$ and takes values in $(0, 1)$, $g(\theta, y)$ is nondecreasing, nonnegative, and measurable. Moreover, $0 < g(\theta, y) \leq 1$, so the boundedness requirement in Assumption 4.3 holds with $G_\theta = 1$.

**Assumption 4.4.** For $a > b$,

$$\frac{g(\theta, a)}{g(\theta, b)} = \left(\frac{\sigma(\alpha a)}{\sigma(\alpha b)}\right)^\theta,$$

which is strictly increasing in $\theta$.

**Fractional exponential:** $g(\theta, y) = \exp\{\theta\, y_+^\alpha\}$, $\alpha > 0$, $y_+ = \max\{y, 0\}$

**Assumption 4.3.** The map $y \mapsto y_+^\alpha$ is nondecreasing and nonnegative, hence $g$ is nondecreasing, nonnegative, and measurable.

**Assumption 4.4.** For $a > b$,

$$\frac{g(\theta, a)}{g(\theta, b)} = \exp\{\theta(a_+^\alpha - b_+^\alpha)\},$$

which is strictly increasing in $\theta$.

**Power–exponential hybrid:** $g(\theta, y) = (y + c)^\beta e^{\theta y}$, $\beta \geq 0$, $c \geq 0$, $y > -c$

**Assumption 4.3.**

$$\partial_y g(\theta, y) = \beta(y + c)^{\beta-1} e^{\theta y} + \theta(y + c)^\beta e^{\theta y} \geq 0,$$

so $g$ is nondecreasing and nonnegative.

**Assumption 4.4.** For $a > b > -c$,

$$\frac{g(\theta, a)}{g(\theta, b)} = \left(\frac{a + c}{b + c}\right)^\beta e^{\theta(a-b)},$$

which is strictly increasing in $\theta$.

**Hyperbolic variant:** $g(\theta, y) = \sinh(\theta(y + c))_+$

Restricting to $y \geq -c$, $g$ is nondecreasing and nonnegative, satisfying Assumption 4.3. For $a > b$, the ratio $\sinh(\theta(a + c))/\sinh(\theta(b + c))$ is strictly increasing in $\theta$, establishing Assumption 4.4.

### Summary

All transformation families above satisfy Assumptions 4.3 and 4.4 under the stated parameter and domain conditions. Exponential and fractional-exponential forms provide strong contrast at the cost of increased variance, while softplus and sigmoid-based transformations offer moderated amplification and improved stability through effective boundedness.

## B. Proofs of Theoretical Results

### B.1. Proof of Theorem 4.1

The proof can be done by following two lemmas.

**Lemma B.1** (CDF gap on a compact window). *Fix $\delta > 0$. Let $p$ be a density that is continuous and strictly positive on an interval $[a, b]$. Let $F$ be its CDF. If $[x - \delta, x + \delta] \subset [a, b]$, then*

$$F(x + \delta) - F(x - \delta) = \int_{x-\delta}^{x+\delta} p(t)\, dt \;\geq\; 2\delta \cdot \inf_{t \in [a,b]} p(t).$$

*Proof.* By definition, $F(x + \delta) - F(x - \delta) = \int_{x-\delta}^{x+\delta} p(t)\, dt$. Since $[x - \delta, x + \delta] \subset [a, b]$, we have $p(t) \geq \inf_{[a,b]} p$ for all $t$ in the integral range. Hence

$$\int_{x-\delta}^{x+\delta} p(t)\, dt \geq \int_{x-\delta}^{x+\delta} \inf_{[a,b]} p\, dt = 2\delta \inf_{[a,b]} p. \tag{22}$$

$\square$

**Lemma B.2** (PDF difference gap away from the mode). *Assume $p \in C^1(\mathbb{R})$ is symmetric and unimodal, and strictly decreasing on $(0, \infty)$. Fix $\delta > 0$ and assume $|c| > r + \delta$. Then there exists $c_\delta(c, r) > 0$ such that*

$$\inf_{|x-c| \leq r} \left| p(x+\delta) - p(x-\delta) \right| \geq c_\delta(c, r). \tag{23}$$

*Proof.* By symmetry it suffices to consider $c > 0$. The condition $c > r + \delta$ implies $x - \delta > 0$ and $x + \delta > 0$ for all $|x - c| \leq r$; hence $x \pm \delta$ lie in the compact interval

$$J := [c - r - \delta, \ c + r + \delta] \subset (0, \infty). \tag{24}$$

Define $\phi(x) := p(x - \delta) - p(x + \delta)$ for $x \in [c - r, c + r]$. Since $p$ is strictly decreasing on $(0, \infty)$ and $x - \delta < x + \delta$, we have $\phi(x) > 0$ for every $x$ in the domain. Moreover, $\phi$ is continuous because $p$ is continuous.

A continuous positive function on a compact set attains a positive minimum. Therefore,

$$m := \min_{x \in [c-r, c+r]} \phi(x) > 0. \tag{25}$$

Finally,

$$|p(x+\delta) - p(x-\delta)| = p(x-\delta) - p(x+\delta) \tag{26}$$

$$= \phi(x) \geq m \ \forall |x - c| \leq r. \tag{27}$$

Setting $c_\delta(c, r) := m$ completes the proof. $\square$

### B.2. Proof of Theorem 4.2

*Proof.* Consider the partial derivative $\partial G_{\theta, \sigma}(\mu) / \partial \mu_i$. We first decompose this derivative into two terms as

$$\frac{\partial G_{\theta, \sigma}(\mu)}{\partial \mu_i} = \int_{\mathbf{x} \in \mathcal{S}} g(\theta, f(\mathbf{x})) \frac{\partial \mathbf{p}_{\mu, \sigma}(\mathbf{x})}{\partial \mu_i} d\mathbf{x} \tag{28}$$

$$= \int_{\mathbf{x} \in \text{cube}(\mathbf{x}^*; \delta)} g(\theta, f(\mathbf{x})) \frac{\partial \mathbf{p}_{\mu, \sigma}(\mathbf{x})}{\partial \mu_i} d\mathbf{x} + \int_{\mathbf{x} \in \mathcal{S} \setminus \text{cube}(\mathbf{x}^*; \delta)} g(\theta, f(\mathbf{x})) \frac{\partial \mathbf{p}_{\mu, \sigma}(\mathbf{x})}{\partial \mu_i} d\mathbf{x} \tag{29}$$

$$:= \frac{\partial H(\mu)}{\partial \mu_i} + \frac{\partial R(\mu)}{\partial \mu_i}, \tag{30}$$

where $\text{cube}(\mathbf{x}^*; \delta) := \{\mathbf{x} | \forall i \in [d], |x_i - x_i^*| \leq \delta\}$ is a $d$ dimensional cube centered at $\mathbf{x}^*$ with $2\delta$ length. Then, the main strategy of the proof is to show that the first term $\frac{\partial H(\mu)}{\partial \mu_i}$ dominates the second term $\frac{\partial R(\mu)}{\partial \mu_i}$ for sufficiently large $\theta$, and then, the sign of $\frac{\partial H(\mu)}{\partial \mu_i}$ behaves like the statement.

First, we bound $\frac{\partial R(\mu)}{\partial \mu_i}$. Let $V_\delta := \sup_{\mathbf{u} \notin \text{cube}(\mathbf{x}^*; \delta)} f(\mathbf{u})$.

$$\left| \frac{\partial R(\mu)}{\partial \mu_i} \right| = \left| \int_{\mathbf{x} \in \mathcal{S} \setminus \text{cube}(\mathbf{x}^*; \delta)} g(\theta, f(\mathbf{x})) \frac{\partial \mathbf{p}_{\mu, \sigma}(\mathbf{x})}{\partial \mu_i} d\mathbf{x} \right| \leq \int_{\mathbf{x} \notin \text{cube}(\mathbf{x}^*; \delta)} g(\theta, f(\mathbf{x})) \left| \frac{\partial \mathbf{p}_{\mu, \sigma}(\mathbf{x})}{\partial \mu_i} \right| d\mathbf{x} \tag{31}$$

$$\leq g(\theta, V_\delta) \int_{\mathbf{x} \notin \text{cube}(\mathbf{x}^*; \delta)} \left| \frac{\partial \mathbf{p}_{\mu, \sigma}(\mathbf{x})}{\partial \mu_i} \right| d\mathbf{x} \leq g(\theta, V_\delta) \int_{\mathbf{x} \in \mathbb{R}^d} \left| \frac{\partial \mathbf{p}_{\mu, \sigma}(\mathbf{x})}{\partial \mu_i} \right| d\mathbf{x} \tag{32}$$

$$= g(\theta, V_\delta) \int_{\mathbf{x}_{\setminus i} \in \mathbb{R}^{d-1}} \prod_{j \neq i}^{d} p\left( \frac{x_j - \mu_j}{\sigma} \right) \frac{d\mathbf{x}_{\setminus i}}{\sigma^{d-1}} \int_{x_i \in \mathbb{R}} \frac{1}{\sigma^2} \left| p'\left( \frac{x_i - \mu_i}{\sigma} \right) \right| dx_i \tag{33}$$

$$= g(\theta, V_\delta) \prod_{j \neq i}^{d} \int_{x_j \in \mathbb{R}} p\left( \frac{x_j - \mu_j}{\sigma} \right) \frac{dx_j}{\sigma} \int_{x_i \in \mathbb{R}} \frac{1}{\sigma^2} \left| p'\left( \frac{x_i - \mu_i}{\sigma} \right) \right| dx_i \tag{34}$$

$$= g(\theta, V_\delta) \int_{z \in \mathbb{R}} \frac{1}{\sigma} \left| p'(z) \right| dz \leq \frac{2g(\theta, V_\delta) p(0)}{\sigma}, \tag{35}$$

where $\mathbf{x}_{\setminus i}$ indicates $d - 1$ dimensional variables except for $x_i$.

Second, we bound $\frac{\partial H(\mu)}{\partial \mu_i}$. Let $D_\delta := (f(\mathbf{x}^*) + V_\delta)/2$. Let $\epsilon_\delta := f(\mathbf{x}^*) - D_\delta$. Then, there exists $\delta'$ such that, for all $x \in \text{cube}(\mathbf{x}^*, \delta')$,

$$f(\mathbf{x}) > f(\mathbf{x}^*) - \epsilon_\delta = D_\delta > V_\delta. \tag{36}$$

Then, under the condition $|\mu_i - x_i^*| > \delta$ in the statement, the sign of the partial derivative does not change in the cube, $\text{cube}(\mathbf{x}^*; \delta)$. In fact, if $\mu_i < x_i^* - \delta$, then for all $\mathbf{x} \in \text{cube}(\mathbf{x}^*; \delta)$, $x_i - \mu_i = x_i - x_i^* + x_i^* - \mu_i > -\delta + \delta = 0$ and $\frac{\partial \mathbf{p}_{\mu,\sigma}(\mathbf{x})}{\partial \mu_i} \geq 0$. If $\mu_i > x_i^* + \delta$, then for all $\mathbf{x} \in \text{cube}(\mathbf{x}^*; \delta)$, $x_i - \mu_i = x_i - x_i^* + x_i^* - \mu_i < \delta - \delta = 0$ and $\frac{\partial \mathbf{p}_{\mu,\sigma}(\mathbf{x})}{\partial \mu_i} \leq 0$. Hence, for $\frac{\partial H(\mu)}{\partial \mu_i}$, we have,

$$\left| \frac{\partial H(\mu)}{\partial \mu_i} \right| = \left| \int_{\mathbf{x} \in \text{cube}(\mathbf{x}^*; \delta)} g(\theta, f(\mathbf{x})) \frac{\partial \mathbf{p}_{\mu,\sigma}(\mathbf{x})}{\partial \mu_i} d\mathbf{x} \right| = \int_{\mathbf{x} \in \text{cube}(\mathbf{x}^*; \delta)} g(\theta, f(\mathbf{x})) \left| \frac{\partial \mathbf{p}_{\mu,\sigma}(\mathbf{x})}{\partial \mu_i} \right| d\mathbf{x} \tag{37}$$

$$\geq \int_{\mathbf{x} \in \text{cube}(\mathbf{x}^*; \delta')} g(\theta, f(\mathbf{x})) \left| \frac{\partial \mathbf{p}_{\mu,\sigma}(\mathbf{x})}{\partial \mu_i} \right| d\mathbf{x} \geq g(\theta, D_\delta) \int_{\mathbf{x} \in \text{cube}(\mathbf{x}^*; \delta')} \left| \frac{\partial \mathbf{p}_{\mu,\sigma}(\mathbf{x})}{\partial \mu_i} \right| d\mathbf{x} \tag{38}$$

$$= g(\theta, D_\delta) \left[ \prod_{j \neq i}^{d} \int_{x_j \in [x_j^* - \delta', x_j^* + \delta']} p\left(\frac{x_j - \mu_j}{\sigma}\right) \frac{dx_j}{\sigma} \right] \int_{x_i \in [x_i^* - \delta', x_i^* + \delta']} \frac{1}{\sigma^2} \left| p'\left(\frac{x_i - \mu_i}{\sigma}\right) \right| dx_i \tag{39}$$

$$= \frac{g(\theta, D_\delta)}{\sigma} \left[ \prod_{j \neq i}^{d} \int_{z \in \left[\frac{x_j^* - \mu_j - \delta'}{\sigma}, \frac{x_j^* - \mu_j + \delta'}{\sigma}\right]} p(z)\, dz \right] \int_{z \in \left[\frac{x_i^* - \mu_i - \delta'}{\sigma}, \frac{x_i^* - \mu_i + \delta'}{\sigma}\right]} \left| p'(z) \right| dz \tag{40}$$

$$= \frac{g(\theta, D_\delta)}{\sigma} \prod_{j \neq i}^{d} \left( F\left(\frac{x_j^* - \mu_j + \delta'}{\sigma}\right) - F\left(\frac{x_j^* - \mu_j - \delta'}{\sigma}\right) \right) \left| p\left(\frac{x_i^* - \mu_i + \delta'}{\sigma}\right) - p\left(\frac{x_i^* - \mu_i - \delta'}{\sigma}\right) \right| \tag{41}$$

$$\geq \frac{g(\theta, D_\delta)}{\sigma} \prod_{j \neq i}^{d} \inf_{z \in [x_j^* + M, x_j^* - M]} \left( F\left(\frac{z + \delta'}{\sigma}\right) - F\left(\frac{z - \delta'}{\sigma}\right) \right) \inf_{z \in [x_i^* + M, x_i^* - M]} \left| p\left(\frac{z + \delta'}{\sigma}\right) - p\left(\frac{z - \delta'}{\sigma}\right) \right| \tag{42}$$

$$\geq \frac{g(\theta, D_\delta)}{\sigma} \cdot \prod_{j \neq i}^{d} C_{\delta'/\sigma}(x_j^*/\sigma, M/\sigma) c_{\delta'/\sigma}(x_i^*/\sigma, M/\sigma) \quad \because \text{Assumption 4.1} \tag{43}$$

Hence, there exists the positive number $\theta$ such that the following inequality holds,

$$\frac{g(\theta, D_\delta)}{\sigma} \cdot \prod_{j \neq i}^{d} C_{\delta'/\sigma}(x_j^*/\sigma, M/\sigma) c_{\delta'/\sigma}(x_i^*/\sigma, M/\sigma) > \frac{g(\theta, V_\delta) \cdot 2p(0)}{\sigma}. \tag{44}$$

Since $D_\delta > V_\delta$ holds and $g(\theta, D_\delta)/g(\theta, V_\delta)$ is increasing due to the Assumption 4.4, we can always pick a sufficiently large $\theta$ that makes the above inequality hold. $\qquad \square$

### B.3. Proof of Theorem 4.3

*Proof.* Under Assumptions 4.1–4.1 and 4.3–4.4, we have the following two properties:

*(1) Lipschitz continuity.* $G_{\theta,\sigma}$ is Lipschitz (hence continuous) on $\mathbb{R}^d$.

$$|G_{\theta,\sigma}(\boldsymbol{\mu}_1) - G_{\theta,\sigma}(\boldsymbol{\mu}_2)| = |\nabla_{\boldsymbol{\mu}} G_{\theta,\sigma}(\nu)^\mathsf{T}(\boldsymbol{\mu}_1 - \boldsymbol{\mu}_2)| \tag{45}$$

$$\leq \|\nabla_{\boldsymbol{\mu}} G_{\theta,\sigma}(\nu)\|_2 \|\boldsymbol{\mu}_1 - \boldsymbol{\mu}_2\|_2 \leq \frac{g_{*,\theta} \sqrt{dI}}{\sigma} \|\boldsymbol{\mu}_1 - \boldsymbol{\mu}_2\|_2 \tag{46}$$

*(2) Vanishing at infinity.* $\lim_{\|\boldsymbol{\mu}\| \to \infty} G_{\theta,\sigma}(\boldsymbol{\mu}) = 0$ and there exists $\boldsymbol{\mu}_0$ with $G_{\theta,\sigma}(\boldsymbol{\mu}_0) > 0$. Since $\mathcal{S} \subset \mathbb{R}^d$ is compact, it is bounded. Assume that $\mathcal{S} \subset \text{cube}(\mathbf{0}, M)$ for some $M > 0$. Then, $g(\theta, f(\mathbf{x})) = 0$ if $\|\mathbf{x}\|_\infty > M$. Hence,

$$G_{\theta,\sigma}(\boldsymbol{\mu}) \leq g_{*,\theta} \prod_{j=1}^{d} \int_{x_j \in [-M, M]} p\left(\frac{x_j - \mu_j}{\sigma}\right) \frac{dx_j}{\sigma} \tag{47}$$

$$= g_{*,\theta} \prod_{j=1}^{d} \left[ F\left(\frac{M - \mu_j}{\sigma}\right) - F\left(\frac{-M - \mu_j}{\sigma}\right) \right] \tag{48}$$

$$\to 0 \text{ as } \|\boldsymbol{\mu}\|_\infty \to \infty, \quad \because \text{For any fixed } \delta, \ \lim_{x \to \pm\infty} F(x) - F(x - \delta) = 0 \tag{49}$$

Given **(1)** and **(2)**, the standard compactness-continuity argument (as in Chen Xu's case) applies verbatim: choose a closed ball where the interior value exceeds the uniform tail bound, invoke the extreme value theorem to attain a maximizer on the ball, and compare with the exterior to conclude global maximality. We omit further details. $\square$

### B.4. Proof of Lemma 4.4

*Proof.* The gradient can be expressed in score form as

$$\nabla_\mu G_{\theta,\sigma}(\mu) = \mathbb{E}\big[g(\theta, f(\mathbf{X})) \nabla_\mu \log \mathbf{p}_{\boldsymbol{\mu},\sigma}(\mathbf{X}); \mathcal{S}\big]. \tag{50}$$

For the $i$th coordinate,

$$\frac{\partial}{\partial \mu_i} \log \mathbf{p}_{\boldsymbol{\mu},\sigma}(\mathbf{x}) = -\frac{1}{\sigma} s(z_i), \tag{51}$$

where $z_i = \frac{x_i - \mu_i}{\sigma}$ and $s(z) = \frac{p'(z)}{p(z)}$. Differentiating once more with respect to $\mu_i$ gives

$$\frac{\partial^2}{\partial^2 \mu_i} \log \mathbf{p}_{\boldsymbol{\mu},\sigma}(\mathbf{x}) = \frac{1}{\sigma^2}\Big(\frac{p''(z_i)}{p(z_i)} - s(z_i)^2\Big). \tag{52}$$

Hence the diagonal entries of the Hessian $H(\boldsymbol{\mu}) = \nabla_\mu^2 G_{\theta,\sigma}(\boldsymbol{\mu})$ are bounded in absolute value by $g_{*,\theta} K / \sigma^2$

$$\nabla_\mu^2 G_{\theta,\sigma}(\boldsymbol{\mu}) = \mathbb{E}\big[g(\theta, f(\mathbf{X})) \nabla_\mu \log \mathbf{p}_{\boldsymbol{\mu},\sigma}(\mathbf{X}) (\nabla_\mu \log \mathbf{p}_{\boldsymbol{\mu},\sigma}(\mathbf{X}))^\top; \mathcal{S}\big] + \mathbb{E}\big[g(\theta, f(\mathbf{X})) \nabla_\mu^2 \log \mathbf{p}_{\boldsymbol{\mu},\sigma}(\mathbf{X}); \mathcal{S}\big]. \tag{53}$$

Hence, $[\nabla_\mu^2 G_{\theta,\sigma}(\boldsymbol{\mu})]_{ii} \le \int |p''(z)| \, dz = K$. It follows that

$$\|H(\mu)\|_{\mathrm{op}} \le \mathrm{tr}\,(H(\mu)) \le \frac{g_{*,\theta} \, d \, K}{\sigma^2}. \tag{54}$$

Finally, the mean value theorem for vector fields implies

$$\|\nabla_\mu G_{\theta,\sigma}(\mu) - \nabla_\mu G_{\theta,\sigma}(\nu)\|_2 \le L_\theta \|\mu - \nu\|_2, \tag{55}$$

with $L_\theta = g_{*,\theta} dK / \sigma^2$, which proves the claim. $\square$

### B.5. Proof of Lemma 4.5

*Proof.* Define $Y = g(\theta, f(X)) \nabla_\mu \log \mathbf{p}(X)$ with mean $m = \mathbb{E}[Y] = \nabla_\mu G_{\theta,\sigma}(\mu)$. For $B$ i.i.d. copies $Y_1, \ldots, Y_B$, independence gives

$$\mathbb{E}\Big\|\tfrac{1}{B} \sum_{k=1}^{B} Y_k\Big\|^2 = \tfrac{1}{B} \mathbb{E}\|Y\|^2 + \tfrac{B-1}{B} \|m\|^2. \tag{56}$$

To bound $\mathbb{E}\|Y\|^2$, note that $\|\nabla_\mu \log \mathbf{p}(X)\|^2 = \sigma^{-2} \sum_{i=1}^d s(Z_i)^2$ with $Z_i = (X_i - \mu_i)/\sigma$ and $s(z) = p'(z)/p(z)$. Taking expectations yields $\mathbb{E}\|\nabla_\mu \log \mathbf{p}(X)\|^2 = (dI)/\sigma^2$. Since $g(\theta, f(X)) \le g_{*,\theta}$, it follows that $\mathbb{E}\|Y\|^2 \le g_{*,\theta}^2 (dI)/\sigma^2$. Moreover, $\|m\|^2 \le \mathbb{E}\|Y\|^2$ by Cauchy-Schwarz, so the same bound holds for $\|m\|^2$. Substituting back into the decomposition gives the uniform bound

$$\mathbb{E}\big[\|\hat{\nabla}_\mu G_{\theta,\sigma}(\mu)\|^2\big] \le g_{*,\theta}^2 \frac{dI}{\sigma^2}. \tag{57}$$

$\square$

### B.6. Proof of Theorem 4.6

*Proof.* By Lemma 4.4, $G_{\theta,\sigma}$ is $L_\theta$-smooth, so for all $\mathbf{x}, \mathbf{y} \in \mathbb{R}^d$,

$$G_{\theta,\sigma}(\mathbf{y}) \ge G_{\theta,\sigma}(\mathbf{x}) + \nabla_\mu G_{\theta,\sigma}(\mathbf{x})^\top (\mathbf{y} - \mathbf{x}) - \frac{L_\theta}{2} \|\mathbf{y} - \mathbf{x}\|^2. \tag{58}$$

Apply (58) with $\mathbf{x} = \boldsymbol{\mu}_t$ and the update $\mathbf{y} = \boldsymbol{\mu}_{t+1} = \boldsymbol{\mu}_t + \eta_t \, \widehat{\nabla}_{\boldsymbol{\mu}} G_{\theta,\sigma}(\boldsymbol{\mu}_t)$ to obtain

$$G_{\theta,\sigma}(\boldsymbol{\mu}_{t+1}) \geq G_{\theta,\sigma}(\boldsymbol{\mu}_t) + \eta_t \, \nabla_{\boldsymbol{\mu}} G_{\theta,\sigma}(\boldsymbol{\mu}_t)^\top \widehat{\nabla}_{\boldsymbol{\mu}} G_{\theta,\sigma}(\boldsymbol{\mu}_t) - \frac{L_\theta \eta_t^2}{2} \left\| \widehat{\nabla}_{\boldsymbol{\mu}} G_{\theta,\sigma}(\boldsymbol{\mu}_t) \right\|^2. \tag{59}$$

Taking conditional expectation given $\boldsymbol{\mu}_t$ and using the score unbiasedness $\mathbb{E}[\widehat{\nabla}_{\boldsymbol{\mu}} G_{\theta,\sigma}(\boldsymbol{\mu}_t) \mid \boldsymbol{\mu}_t] = \nabla_{\boldsymbol{\mu}} G_{\theta,\sigma}(\boldsymbol{\mu}_t)$ converts the mixed inner product into $\eta_t \| \nabla_{\boldsymbol{\mu}} G_{\theta,\sigma}(\boldsymbol{\mu}_t) \|^2$. Taking total expectation and invoking the uniform bound from Lemma 4.5 yields

$$\mathbb{E}[G_{\theta,\sigma}(\boldsymbol{\mu}_{t+1})] - \mathbb{E}[G_{\theta,\sigma}(\boldsymbol{\mu}_t)] \geq \eta_t \, \mathbb{E}[\| \nabla_{\boldsymbol{\mu}} G_{\theta,\sigma}(\boldsymbol{\mu}_t) \|^2] - \frac{L_\theta \eta_t^2}{2} \, Q_\theta. \tag{60}$$

Summing (60) over $t = 0, \ldots, T-1$ telescopes the left-hand side to $\mathbb{E}[G_{\theta,\sigma}(\boldsymbol{\mu}_T)] - \mathbb{E}[G_{\theta,\sigma}(\boldsymbol{\mu}_0)]$. Hence

$$\sum_{t=0}^{T-1} \eta_t \, \mathbb{E}[\| \nabla_{\boldsymbol{\mu}} G_{\theta,\sigma}(\boldsymbol{\mu}_t) \|^2] \leq \mathbb{E}[G_{\theta,\sigma}(\boldsymbol{\mu}_T)] - \mathbb{E}[G_{\theta,\sigma}(\boldsymbol{\mu}_0)] + \frac{L_\theta Q_\theta}{2} \sum_{t=0}^{T-1} \eta_t^2. \tag{61}$$

$\square$

### B.7. Proof of Corollary 4.7

*Proof.* The localization claim follows from Theorem 4.2: for each $\delta > 0$ and $M > 0$ there exists $\theta_{\delta,\sigma,M}$ such that the coordinatewise derivatives have fixed signs outside $\mathrm{cube}(\mathbf{x}^*; \delta)$ within $\{\|\boldsymbol{\mu}\|_\infty < M\}$, hence no stationary point (and therefore no local maximizer) can lie outside $\mathrm{cube}(\mathbf{x}^*; \delta)$.

For the rate, start from Theorem 4.6. With the polynomial steps,

$$\sum_{t=0}^{\infty} \eta_t^2 \leq \sigma^4 \left( 1 + \frac{1}{2\gamma} \right), \quad \sum_{t=0}^{T-1} \eta_t \geq \frac{\sigma^2 (2^{\frac{1}{2} - \gamma} - 1)}{\frac{1}{2} - \gamma} \, T^{\frac{1}{2} - \gamma}. \tag{62}$$

From $G_\theta \leq (dI/\sigma^2) \, g_{*,\theta}^2$ and $L_\theta \leq g_{*,\theta} K / \sigma^2$, we have

$$\min_{0 \leq t \leq T-1} \mathbb{E}[\| \nabla_{\boldsymbol{\mu}} G_{\theta,\sigma}(\boldsymbol{\mu}_t) \|^2] \leq \frac{g_{*,\theta} + \frac{1}{2} L_\theta G_\theta \sum_t \eta_t^2}{\sum_t \eta_t} \leq \frac{g_{*,\theta} + d \, IK \, g_{*,\theta}^3}{\sigma^2 \, T^{\frac{1}{2} - \gamma}} \cdot C_\gamma, \tag{63}$$

where $C_\gamma := (\frac{1}{2} - \gamma)/(2^{\frac{1}{2} - \gamma} - 1)$ which implies the stated $T$ bound after inversion. Finally, we get the iteration complexity $T > \left( C_\gamma \cdot \frac{g_{*,\theta} + d \, IK \, g_{*,\theta}^3}{\sigma^2 \varepsilon} \right)^{\frac{2}{1 - 2\gamma}}$. $\square$

### B.8. Proof of Corollary 4.8

*Proof.* We work in the single-kernel, anisotropic setting $\mathbf{p}_{\mu,\Sigma}(x) = \prod_{i=1}^{d} \sigma_i^{-1} p\big((x_i - \mu_i)/\sigma_i\big)$. It is enough to derive the Lipschitz constant and the second moment bound. For the Lipschitz constant, let $H(\mu) := \nabla_\mu^2 G_{\theta,\Sigma}(\mu)$ and use the induced 2-operator norm. We used $\|H(\mu)\|_2 \leq \mathrm{tr}(H(\mu)) \leq g_{*,\theta} K \sum_i \sigma_i^{-2}$. Thus,

$$L_\theta(\Sigma) \leq g_{*,\theta} K \sum_{i=1}^{d} \sigma_i^{-2}. \tag{64}$$

The second moment bound can be obtained as

$$\mathbb{E}[\| \hat{\nabla}_\mu G_{\theta,\sigma}(\mu) \|^2] \leq \mathbb{E}[\| g(\theta, f(X)) \nabla_\mu \log \mathbf{p}(X) \|^2] \leq g_{*,\theta}^2 \, I \, S_2(\Sigma). \tag{65}$$

Hence, $Q_\theta(\Sigma)$ is $g_{*,\theta}^2 \, I \, S_2(\Sigma)$. Then, remaining parts of the proof is similar to Corollary 4.7. $\square$

### B.9. Proof of Lemma 4.9

*Proof.* Given $\boldsymbol{\mu}_t$, $b_k$ is a function of $\{(\mathbf{X}^{(j)})\}_{j \neq k}$ and thus independent of $\mathbf{X}^{(k)}$. Using $\mathbb{E}[S(\mathbf{X}^{(k)}) \mid \boldsymbol{\mu}_t] = 0$,

$$\mathbb{E}\Big[ \big(h(\mathbf{X}^{(k)}) - b_k\big) S(\mathbf{X}^{(k)}) \mid \boldsymbol{\mu}_t \Big] = \mathbb{E}\Big[ h(\mathbf{X}^{(k)}) S(\mathbf{X}^{(k)}) \mid \boldsymbol{\mu}_t \Big] - \mathbb{E}\Big[ b_k S(\mathbf{X}^{(k)}) \mid \boldsymbol{\mu}_t \Big] \tag{66}$$

$$= \mathbb{E}\Big[h(\mathbf{X}^{(k)})\,S(\mathbf{X}^{(k)}) \mid \boldsymbol{\mu}_t\Big] - \mathbb{E}[b_k \mid \boldsymbol{\mu}_t]\,\underbrace{\mathbb{E}\Big[S(\mathbf{X}^{(k)}) \mid \boldsymbol{\mu}_t\Big]}_{=\,0} \tag{67}$$

$$= \mathbb{E}\Big[h(\mathbf{X}^{(k)})\,S(\mathbf{X}^{(k)}) \mid \boldsymbol{\mu}_t\Big]. \tag{68}$$

Averaging over $k$ yields the claim. □

### B.10. Proof of Lemma 4.10

*Proof.* Let $Y_k = (h(\mathbf{X}^{(k)}) - b_k)\,S(\mathbf{X}^{(k)})$. Then,

$$\mathbb{E}\Big[\big\|\tfrac{1}{B}\sum_k Y_k\big\|^2\Big|\boldsymbol{\mu}_t\Big] \le \frac{1}{B}\sum_{k=1}^{B}\mathbb{E}\big[\|Y_k\|^2\big|\boldsymbol{\mu}_t\big] = \mathbb{E}\big[\|Y_1\|^2\big|\boldsymbol{\mu}_t\big].$$

Since $\|Y_1\|^2 = (h(\mathbf{X}^{(1)}) - b_1)^2\,\|S(\mathbf{X}^{(1)})\|^2$, we introduce an auxiliary constant baseline $b^\star$ and write

$$h(\mathbf{X}^{(1)}) - b_1 = \big(h(\mathbf{X}^{(1)}) - b^\star\big) + \big(b^\star - b_1\big).$$

Expanding and taking conditional expectation yields

$$\mathbb{E}\big[\|Y_1\|^2 \mid \boldsymbol{\mu}_t\big] = \underbrace{\mathbb{E}\Big[(h(\mathbf{X}^{(1)}) - b^\star)^2\|S(\mathbf{X}^{(1)})\|^2\Big|\boldsymbol{\mu}_t\Big]}_{=:A} + \underbrace{\mathbb{E}\Big[(b^\star - b_1)^2\|S(\mathbf{X}^{(1)})\|^2\Big|\boldsymbol{\mu}_t\Big]}_{=:B}$$
$$+ 2\mathbb{E}\Big[(b^\star - b_1)(h(\mathbf{X}^{(1)}) - b^\star)\|S(\mathbf{X}^{(1)})\|^2 \Big| \boldsymbol{\mu}_t\Big].$$

By construction, $b_1$ is independent of $\mathbf{X}^{(1)}$, and hence independent of $(h(\mathbf{X}^{(1)}) - b^\star)\|S(\mathbf{X}^{(1)})\|^2$. Moreover, $b^\star$ is defined as the optimal constant baseline minimizing $\mathbb{E}[(h(\mathbf{X}) - b)^2\|S(\mathbf{X})\|^2|\mu_t]$, which implies the first-order optimality condition

$$\mathbb{E}\Big[(h(\mathbf{X}^{(1)}) - b^\star)\|S(\mathbf{X}^{(1)})\|^2|\mu_t\Big] = 0.$$

Therefore,

$$\mathbb{E}\Big[(b^\star - b_1)(h(\mathbf{X}^{(1)}) - b^\star)\|S(\mathbf{X}^{(1)})\|^2|\mu_t\Big] = 0,$$

and the final term vanishes after taking conditional expectation. Consequently, it suffices to analyze the two terms $A$ and $B$ and derive explicit upper bounds for each. For the first term $A$, by the definition of $R^2_{\theta,\mu,\Sigma}$ and $g(\theta,f) \le g_{*,\theta}$,

$$A = (1 - R^2_{\theta,\boldsymbol{\mu}_t,\Sigma})\,\mathbb{E}[h(\mathbf{X})^2\|S(\mathbf{X})\|^2] \le (1 - R^2_{\theta,\boldsymbol{\mu}_t,\Sigma})\,g^2_{*,\theta}\,I\,S_2(\Sigma). \tag{69}$$

For the second term $B$, note that $b_k$ is independent of $\mathbf{X}^{(k)}$ conditional on $\boldsymbol{\mu}_t$, so

$$\mathbb{E}[(b^\star - b_k)^2\|S(\mathbf{X})\|^2 \mid \boldsymbol{\mu}_t] = \mathbb{E}[(b^\star - b_k)^2 \mid \boldsymbol{\mu}_t]\,\mathbb{E}[\|S(\mathbf{X})\|^2 \mid \boldsymbol{\mu}_t] \le \mathbb{E}[(b^\star - b_k)^2]\,I\,S_2(\Sigma). \tag{70}$$

We now bound $\mathbb{E}[(b^\star - b_k)^2]$ using the mean value theorem for the two-variable map $g(u,v) = u/(v + \lambda)$. Decompose

$$b^\star - b_k = \Big(\tfrac{\mu_U}{\mu_V} - \tfrac{\mu_U}{\mu_V + \lambda}\Big) + \Big(\tfrac{\mu_U}{\mu_V + \lambda} - \tfrac{U_{\backslash k}}{V_{\backslash k} + \lambda}\Big) =: \delta_0 - \Delta, \tag{71}$$

so that $\delta_0 = \frac{b^\star \lambda}{\mu_V + \lambda}$ captures the deterministic ridge bias and $\Delta$ is the random fluctuation around $(\mu_U, \mu_V)$.

Let $U_t = \mu_U + t\Delta_u$ and $V_t = \mu_V + t\Delta_v$ with $\Delta_u = U_{\backslash k} - \mu_U$ and $\Delta_v = V_{\backslash k} - \mu_V$. By the (integral) mean value representation,

$$\Delta = g(U_{\backslash k}, V_{\backslash k}) - g(\mu_U, \mu_V) = \int_0^1 \nabla g(U_t, V_t) \cdot (\Delta_u, \Delta_v)\, dt. \tag{72}$$

Since $\nabla g(u,v) = \big((v + \lambda)^{-1},\, -u(v + \lambda)^{-2}\big)$, it follows that

$$|\Delta| \le \int_0^1 \Big(\frac{|\Delta_u|}{V_t + \lambda} + \frac{|U_t|\,|\Delta_v|}{(V_t + \lambda)^2}\Big)\, dt \le \frac{|\Delta_u|}{\lambda} + \frac{(\mu_U + |\Delta_u|)\,|\Delta_v|}{\lambda^2}, \tag{73}$$

where we used $V_t + \lambda \geq \lambda$ and $|U_t| \leq \mu_U + |\Delta_u|$.

Squaring and using $(x + y)^2 \leq 2x^2 + 2y^2$ gives

$$\Delta^2 \leq \frac{2\,\Delta_u^2}{\lambda^2} + \frac{2(\mu_U + |\Delta_u|)^2\,\Delta_v^2}{\lambda^4} \leq \frac{2\,\Delta_u^2}{\lambda^2} + \frac{4\mu_U^2\,\Delta_v^2}{\lambda^4} + \frac{4\,\Delta_u^2\,\Delta_v^2}{\lambda^4}. \tag{74}$$

Taking expectations and using the sample-mean scalings $\mathbb{E}[\Delta_u^2] = \mathrm{Var}(U)/s$, $\mathbb{E}[\Delta_v^2] = \mathrm{Var}(V)/s$, and $\mathbb{E}[\Delta_u^2\Delta_v^2] \leq (\mathbb{E}\Delta_u^4)^{1/2}(\mathbb{E}\Delta_v^4)^{1/2} = O(s^{-2})$, we obtain

$$\mathbb{E}[\Delta^2] \leq \frac{2}{\lambda^2} \cdot \frac{\mathrm{Var}(U)}{s} + \frac{4\mu_U^2}{\lambda^4} \cdot \frac{\mathrm{Var}(V)}{s} + O\Big(\frac{1}{\lambda^4 s^2}\Big). \tag{75}$$

Finally,

$$\mathbb{E}[(b^\star - b_k)^2] = \delta_0^2 - 2\delta_0\mathbb{E}[\Delta] + \mathbb{E}[\Delta^2] \leq \left(\frac{b^\star \lambda}{\mu_V + \lambda}\right)^2 + \frac{2\,\mathrm{Std}(U)}{\lambda\sqrt{s}} + \frac{2\mu_U\,\mathrm{Std}(V)}{\lambda^2\sqrt{s}} + O\left(\frac{1}{\lambda^2 s}\right)$$
$$+ \frac{2\,\mathrm{Var}(U)}{\lambda^2 s} + \frac{4\mu_U^2\,\mathrm{Var}(V)}{\lambda^4 s} + O\Big(\frac{1}{\lambda^4 s^2}\Big), \tag{76}$$

which is a mean-value-theorem alternative to the Taylor-based bound and has the same rate profile in $s = B - 1$. $\qquad\square$

## B.11. Proof of Theorem 4.11

*Proof.* The claim follows directly from Lemma 4.10 with an appropriate choice of the ridge parameter. From (17), for any $\lambda > 0$ we have

$$\mathbb{E}\left[\left\|\widehat{\nabla}_{\boldsymbol{\mu}}^{\mathrm{loo}} G_{\theta,\Sigma}(\boldsymbol{\mu}_t)\right\|^2 \Big| \boldsymbol{\mu}_t\right] \leq \left((1 - R_{\theta,\boldsymbol{\mu}_t,\Sigma}^2)\, g_{*,\theta}^2 + C_{\mathrm{loo}}(B, \lambda)\right) I\, S_2(\Sigma). \tag{77}$$

We choose $\lambda = (B - 1)^{-1/8}$. Substituting this choice into the expression of $C_{\mathrm{loo}}(B, \lambda)$ in (18) shows that each term in $C_{\mathrm{loo}}(B, \lambda)$ decays at rate $O((B-1)^{-1/4})$, and therefore $C_{\mathrm{loo}}(B, \lambda) = O(B^{-1/4})$. This yields the first inequality stated in the corollary.

Taking expectation of (77) with respect to $\boldsymbol{\mu}_t$ gives

$$\mathbb{E}\left[\left\|\widehat{\nabla}_{\boldsymbol{\mu}}^{\mathrm{loo}} G_{\theta,\Sigma}(\boldsymbol{\mu}_t)\right\|^2\right] \leq \left(g_{*,\theta}^2\big(1 - \mathbb{E}[R_{\theta,\boldsymbol{\mu}_t,\Sigma}^2]\big) + C_{\mathrm{loo}}(B, \lambda)\right) I\, S_2(\Sigma). \tag{78}$$

By definition of $C_{\theta,\Sigma}$, we have $\mathbb{E}[R_{\theta,\boldsymbol{\mu}_t,\Sigma}^2] \geq C_{\theta,\Sigma}$ for all $t$, and hence

$$\mathbb{E}\left[\left\|\widehat{\nabla}_{\boldsymbol{\mu}}^{\mathrm{loo}} G_{\theta,\Sigma}(\boldsymbol{\mu}_t)\right\|^2\right] \leq \left(g_{*,\theta}^2(1 - C_{\theta,\Sigma}) + C_{\mathrm{loo}}(B, \lambda)\right) I\, S_2(\Sigma). \tag{79}$$

Since $C_{\mathrm{loo}}(B, \lambda) = O(B^{-1/4})$, there exists a numerical constant $c > 0$ such that $C_{\mathrm{loo}}(B, \lambda) \leq c\, B^{-1/4}$. Under the assumption $B = \Omega\big(16/(C_{\theta,\Sigma} g_{*,\theta}^2)^4\big)$, the remainder term satisfies $c\, B^{-1/4} \leq \frac{1}{2} C_{\theta,\Sigma} g_{*,\theta}^2$. Substituting this bound into the previous inequality yields

$$\mathbb{E}\left[\left\|\widehat{\nabla}_{\boldsymbol{\mu}}^{\mathrm{loo}} G_{\theta,\Sigma}(\boldsymbol{\mu}_t)\right\|^2\right] \leq \left(1 - \frac{C_{\theta,\Sigma}}{2}\right) g_{*,\theta}^2\, I\, S_2(\Sigma), \tag{80}$$

which proves the claim. $\qquad\square$

## B.12. Proof of Corollary 4.12

*Proof.* The one-step expected improvement inequality in Theorem 4.6 holds with the same smoothness constant $L_\theta(\Sigma) = g_{*,\theta} K\, S_2(\Sigma)$ from Lemma 4.4. Replacing the plain score second moment $Q_\theta(\Sigma) = g_{*,\theta}^2 I S_2(\Sigma)$ by the LOO constant from Theorem 4.11, we have

$$Q_\theta^{(\mathrm{loo})}(\Sigma) = \left((1 - \tfrac{C_{\theta,\Sigma}}{2}) + O(B^{-1/4})\right) g_{*,\theta}^2\, I\, S_2(\Sigma).$$

Plugging $L_\theta(\Sigma)$ and $Q_\theta^{(\mathrm{loo})}(\Sigma)$ into the telescoping argument of Theorem 4.6, and then using the polynomial step schedule $\eta_t = S_2(\Sigma)^{-1}(t + 1)^{-(\frac{1}{2} + \gamma)}$ (whose sums satisfy $\sum_{t=0}^{T-1} \eta_t \geq C_\gamma S_2(\Sigma)^{-1} T^{\frac{1}{2} - \gamma}$ and $\sum_{t=0}^{\infty} \eta_t^2 \leq (1 + \frac{1}{2\gamma}) S_2(\Sigma)^{-2})$. $\qquad\square$

# C. Experimental Detail

This appendix provides full experimental details for the canonical optimization benchmarks and black-box targeted adversarial attack experiments reported in the main text, including hyperparameter selection procedures, optimization settings, and dataset-specific configurations.

## C.1. Canonical Benchmark Objective Functions

**Ackley.**

$$\text{Ackley}(\mathbf{x}) = 20 \exp\left(-0.2\sqrt{\frac{1}{D}\sum_{d=1}^{D} x_d^2}\right) + \exp\left(\frac{1}{D}\sum_{d=1}^{D} \cos(2\pi x_d)\right) - 20 - e, \tag{81}$$

which attains its global optimum value of $0$ at $\mathbf{x} = \mathbf{0}$.

**Rosenbrock.**

$$\text{Rosenbrock}(\mathbf{x}) = -\frac{1}{D-1}\sum_{d=1}^{D-1}\left[100(x_{d+1} - x_d^2)^2 + (1 - x_d)^2\right], \tag{82}$$

which attains its global optimum value of $0$ at $\mathbf{x} = \mathbf{1}$.

**Griewank.**

$$\text{Griewank}(\mathbf{x}) = -1 - \frac{1}{4000}\sum_{d=1}^{D} x_d^2 + \prod_{d=1}^{D} 1.05^{0.2}\cos\left(\frac{x_d}{\sqrt{d}}\right), \tag{83}$$

which attains its global optimum value of $1.05^{0.2D} - 1$ at $\mathbf{x} = \mathbf{0}$.

## C.2. Hyperparameters of Canonical Benchmarks

Hyperparameters for the benchmark test functions were determined by averaging results from 20 trajectories initialized at different points. For each configuration, we calculated the average of the best MSE values obtained across these 20 runs and selected the one with the lowest overall mean. These selected values are reported in the table 9, 10, and 11. The experimental environment was fixed with dimension $D = 500$, the total number of optimization steps per run was set to $T = 400$, and the number of Monte Carlo samples for gradient estimation was set to $B = 50$.

*Table 9.* Hyperparameters for Optimizing Ackley. The initial points for the Ackley function were sampled from a multivariate normal distribution $\mathcal{N}(5\mathbf{1}, 0.01^2\mathbf{I})$. The candidate set for initial learning rate ($\eta_0$) is $\mathcal{L} := \{0.1, 0.5\}$, and for initial smoothing scale ($\sigma$) is $\mathcal{S} := \{0.1, 0.5\}$. The candidate set for the amplification parameter ($\theta$) is $\mathcal{A} := \{1, 3, 5\}$. For both ProMoT and ProMoT-loo, a Logistic kernel was employed as the smoothing kernel, with the transformation function defined as $g(\theta, y) = (y + c)^\beta e^{\theta y}$ (where $c = 600$ and $\beta = 10$). In RSGF and ZO-SLGHd/r, $\gamma$ denotes the decreasing factor for the smoothing scale $\sigma$, while in the baseline ZO-SLGHd, $\alpha$ represents the step size used to update the smoothing scale.

| Method | Selected Values | Candidates |
|---|---|---|
| ProMoT | $\eta_0 = 0.5, \sigma = 0.5, \theta = 5$ | $\eta_0 \in \mathcal{L}, \sigma \in \mathcal{S}, \theta \in \mathcal{A}$ |
| ProMoT-loo | $\eta_0 = 0.5, \sigma = 0.1, \theta = 1$ | $\eta_0 \in \mathcal{L}, \sigma \in \mathcal{S}, \theta \in \mathcal{A}$ |
| EPGS | $\eta_0 = 0.5, \sigma = 0.5, \theta = 5$ | $\eta_0 \in \mathcal{L}, \sigma \in \mathcal{S}, \theta \in \mathcal{A}$ |
| RSGF | $\eta_0 = 0.1, \sigma = 0.1, \gamma = 0.9$ | $\eta_0 \in \mathcal{L}, \sigma \in \mathcal{S}, \gamma \in \{0.8, 0.9, 0.99\}$ |
| ZO-SGD | $\eta_0 = 10, \sigma = 0.1$ | $\eta_0 \in \{1, 5, 10\}, \sigma \in \mathcal{S}$ |
| ZO-AdaMM | $\eta_0 = 0.5, \sigma = 0.1, \beta_1 = 0.5, \beta_2 = 0.5$ | $\eta_0 \in \mathcal{L}, \sigma \in \mathcal{S}, \beta_1 \in \{0.5, 0.7, 0.9\}, \beta_2 \in \{0.1, 0.3, 0.5\}$ |
| ZO-SLGHd | $\eta_0 = 5, \sigma = 0.5, \gamma = 0.99, \alpha = 0.001$ | $\eta_0 \in \{1, 5, 10\}, \sigma \in \mathcal{S}, \gamma \in \{0.95, 0.99\}, \alpha \in \{0.1, 0.01, 0.001\}$ |
| ZO-SLGHr | $\eta_0 = 5, \sigma = 0.5, \gamma = 0.99$ | $\eta_0 \in \{1, 5, 10\}, \sigma \in \mathcal{S}, \gamma \in \{0.95, 0.99\}$ |
| CMA-ES | $\sigma = 0.5$ | $\sigma \in \mathcal{S}$ |

## C.3. Experimental Settings for Black-Box Targeted Adversarial Attacks

We first describe the black-box attack objective shared across all adversarial experiments, and then provide dataset-specific model architectures, preprocessing procedures, and optimization hyperparameters.

*Table 10.* Hyperparameters for Optimizing Rosenbrock. The initial points for the Rosenbrock function were sampled from a multivariate normal distribution $\mathcal{N}(3\mathbf{1}, 0.01^2\mathbf{I})$. The candidate set for initial learning rate ($\eta_0$) is $\mathcal{L} := \{0.1, 0.5\}$, and for initial smoothing scale ($\sigma$) is $\mathcal{S} := \{0.1, 0.5\}$. The candidate set for the amplification parameter ($\theta$) is $\mathcal{A} := \{0.1, 0.01, 0.001\}$. For both ProMoT and ProMoT-loo, a Logistic kernel was employed as the smoothing kernel, with the transformation function defined as $g(\theta, y) = (y + c)^\beta e^{\theta y}$ (where $c = 6,000$ and $\beta = 10$). In RSGF and ZO-SLGHd/r, $\gamma$ denotes the decreasing factor for the smoothing scale $\sigma$, while in the baseline ZO-SLGHd, $\alpha$ represents the step size used to update the smoothing scale.

| Method | Selected Values | Candidates |
|---|---|---|
| ProMoT | $\eta_0 = 0.1, \sigma = 0.1, \theta = 0.1$ | $\eta_0 \in \mathcal{L}, \sigma \in \mathcal{S}, \theta \in \mathcal{A}$ |
| ProMoT-loo | $\eta_0 = 0.1, \sigma = 0.1, \theta = 0.001$ | $\eta_0 \in \mathcal{L}, \sigma \in \mathcal{S}, \theta \in \mathcal{A}$ |
| EPGS | $\eta_0 = 0.1, \sigma = 0.1, \theta = 0.1$ | $\eta_0 \in \mathcal{L}, \sigma \in \mathcal{S}, \theta \in \mathcal{A}$ |
| RSGF | $\eta_0 = 0.1, \sigma = 0.5, \gamma = 0.99$ | $\eta_0 \in \mathcal{L}, \sigma \in \mathcal{S}, \gamma \in \{0.9, 0.95, 0.99\}$ |
| ZO-SGD | $\eta_0 = 0.01, \sigma = 0.1$ | $\eta_0 \in \{0.01, 0.001\}, \sigma \in \mathcal{S}$ |
| ZO-AdaMM | $\eta_0 = 0.5, \sigma = 0.1, \beta_1 = 0.3, \beta_2 = 0.9$ | $\eta_0 \in \mathcal{L}, \sigma \in \mathcal{S}, \beta_1 \in \{0.5, 0.7, 0.9\}, \beta_2 \in \{0.1, 0.3, 0.5\}$ |
| ZO-SLGHd | $\eta_0 = 0.001, \sigma = 0.5, \gamma = 0.95, \alpha = 0.1$ | $\eta_0 \in \{0.01, 0.001\}, \sigma \in \mathcal{S}, \gamma \in \{0.95, 0.99\}, \alpha \in \{0.1, 0.01, 0.001\}$ |
| ZO-SLGHr | $\eta_0 = 0.001, \sigma = 0.5, \gamma = 0.95$ | $\eta_0 \in \{0.01, 0.001\}, \sigma \in \mathcal{S}, \gamma \in \{0.95, 0.99\}$ |
| CMA-ES | $\sigma = 0.1$ | $\sigma \in \mathcal{S}$ |

*Table 11.* Hyperparameters for Optimizing Griewank. The initial points for the Griewank function were sampled from a multivariate normal distribution $\mathcal{N}(5\mathbf{1}, 0.01^2\mathbf{I})$. The candidate set for initial learning rate ($\eta_0$) is $\mathcal{L} := \{0.1, 0.5\}$, and for initial smoothing scale ($\sigma$) is $\mathcal{S} := \{1, 2\}$. The candidate set for the amplification parameter ($\theta$) is $\mathcal{A} := \{1, 3, 5\}$. For both ProMoT and ProMoT-loo, a Logistic kernel was employed as the smoothing kernel, with the transformation function defined as $g(\theta, y) = (y + c)^\beta e^{\theta y}$ (where $c = 1,000$ and $\beta = 10$). In RSGF and ZO-SLGHd/r, $\gamma$ denotes the decreasing factor for the smoothing scale $\sigma$, while in the baseline ZO-SLGHd, $\alpha$ represents the step size used to update the smoothing scale.

| Method | Selected Values | Candidates |
|---|---|---|
| ProMoT | $\eta_0 = 0.1, \sigma = 2, \theta = 5$ | $\eta_0 \in \mathcal{L}, \sigma \in \mathcal{S}, \theta \in \mathcal{A}$ |
| ProMoT-loo | $\eta_0 = 0.1, \sigma = 1, \theta = 1$ | $\eta_0 \in \mathcal{L}, \sigma \in \mathcal{S}, \theta \in \mathcal{A}$ |
| EPGS | $\eta_0 = 0.1, \sigma = 2, \theta = 5$ | $\eta_0 \in \mathcal{L}, \sigma \in \mathcal{S}, \theta \in \mathcal{A}$ |
| RSGF | $\eta_0 = 0.1, \sigma = 1, \gamma = 0.9$ | $\eta_0 \in \mathcal{L}, \sigma \in \mathcal{S}, \gamma \in \{0.8, 0.9, 0.99\}$ |
| ZO-SGD | $\eta_0 = 10, \sigma = 1$ | $\eta_0 \in \{1, 5, 10\}, \sigma \in \mathcal{S}$ |
| ZO-AdaMM | $\eta_0 = 0.5, \sigma = 1, \beta_1 = 0.1, \beta_2 = 0.7$ | $\eta_0 \in \mathcal{L}, \sigma \in \mathcal{S}, \beta_1 \in \{0.5, 0.7, 0.9\}, \beta_2 \in \{0.1, 0.3, 0.5\}$ |
| ZO-SLGHd | $\eta_0 = 10, \sigma = 1, \gamma = 0.99, \alpha = 0.1$ | $\eta_0 \in \{1, 5, 10\}, \sigma \in \mathcal{S}, \gamma \in \{0.95, 0.99\}, \alpha \in \{0.1, 0.01, 0.001\}$ |
| ZO-SLGHr | $\eta_0 = 10, \sigma = 2, \gamma = 0.95$ | $\eta_0 \in \{1, 5, 10\}, \sigma \in \mathcal{S}, \gamma \in \{0.95, 0.99\}$ |
| CMA-ES | $\sigma = 2$ | $\sigma \in \mathcal{S}$ |

**Attack objective.** Let $C$ denote a black-box classifier and $\mathbf{x}$ an input sample. A targeted adversarial attack seeks a perturbation $\boldsymbol{\mu}$ such that

$$\arg\max_y C(\mathbf{x} + \boldsymbol{\mu})_y = y_{\text{tgt}},$$

where the target label is chosen as the *most unlikely class* $y_{\text{tgt}} = \arg\min_y C(\mathbf{x})_y$. We optimize the C&W-style black-box objective (Carlini & Wagner, 2017; Xu, 2025)

$$L(\boldsymbol{\mu}) = -\max\left\{ \max_{y \neq y_{\text{tgt}}} C(\mathbf{x} + \boldsymbol{\mu})_y - C(\mathbf{x} + \boldsymbol{\mu})_{y_{\text{tgt}}}, \kappa \right\} - \lambda \|\boldsymbol{\mu}\|_2, \tag{84}$$

which encourages the target logit to exceed all non-target logits by a margin $\kappa$ while penalizing the perturbation magnitude. Once the margin constraint is satisfied, optimization focuses on minimizing $\|\boldsymbol{\mu}\|_2$, leading to less perceptible adversarial examples.

**CIFAR-10.** We evaluated the proposed methods and baselines against a CNN-based classifier trained on the CIFAR-10 dataset (Krizhevsky et al., 2009). Specifically, the target model's architecture, following the configurations of the model in Tables 1 and 2 of (Carlini & Wagner, 2017), was implemented using the PyTorch framework. To evaluate the effectiveness of our methods and baselines against a robust target, defensive distillation (Papernot et al., 2016) was applied during the training process. After training, the model achieved a test accuracy of 75.7%, serving as a representative target for our adversarial attack evaluations. Detailed hyperparameter configurations for these experiments are listed in Table 12. The input dimension was set to $D = 3,072$ (corresponding to $32 \times 32 \times 3$ pixels), and we limited the total number of optimization steps per run to $T = 500$. For gradient estimation, we employed $B = 30$ Monte Carlo samples.

*Table 12.* Hyperparameters for CIFAR-10 Attack. In this experiment, $\eta_0$, $\sigma$, and $\theta$ denote the initial learning rate, the initial smoothing scale, and the amplification parameter, respectively. For the smoothing kernel, ProMoT employed a Gaussian kernel, while ProMoT-loo utilized a Generalized Gaussian kernel. Both algorithms shared the same transformation function, defined as $g(\theta, y) = (y + c)^\beta e^{\theta y}$ (where $c = 10,000$ and $\beta = 10$). Following the baseline configurations, $\gamma$ represents the decreasing factor for the smoothing scale $\sigma$ in RSGF and ZO-SLGHd/r, and $\alpha$ denotes the step size used for updating the smoothing scale in ZO-SLGHd.

| Method | Hyperparameters |
|---|---|
| ProMoT | $\eta_0 = 0.005, \sigma = 0.1, \theta = 0.03$ |
| ProMoT-loo | $\eta_0 = 0.005, \sigma = 0.01, \theta = 0.03$ |
| EPGS | $\eta_0 = 0.005, \sigma = 0.1, \theta = 0.03$ |
| RSGF | $\eta_0 = 0.03, \sigma = 1.0, \gamma = 0.8$ |
| ZO-SGD | $\eta_0 = 0.00005, \sigma = 0.1$ |
| ZO-AdaMM | $\eta_0 = 0.03, \sigma = 0.1, \beta_1 = 0.9, \beta_2 = 0.1$ |
| ZO-SLGHd | $\eta_0 = 0.00003, \sigma = 0.1, \gamma = 0.999, \alpha = 0.1/3072$ |
| ZO-SLGHr | $\eta_0 = 0.00003, \sigma = 0.1, \gamma = 0.995$ |
| CMA-ES | $\sigma = 0.05$ |

**VitalDB.** We used the case-level clinical table (cases) from the VitalDB dataset (Lee et al., 2022) and defined `death_inhosp` as the label. We binarized `sex` and removed categorical fields as well as metadata identifiers and timestamps. Missing values were imputed as follows: (i) `intraop_ebl`, `intraop_uo`, and `intraop_crystalloid` were set to 0; (ii) arterial blood gas variables were filled with normal reference values; and (iii) all remaining features were imputed using the feature-wise median. We applied a $\log(1 + x)$ transform to skewed lab and medication variables and applied a Yeo–Johnson transform to `preop_be`. Finally, min–max scaling was used to map all features to $[-2, 2]$. This resulted in a final input dimension of $D = 42$. All preprocessing transforms were fitted on the training split only and then applied to the test split.

The XGBoost classifier was trained on a highly imbalanced dataset consisting of 4,791 training samples and 1,597 test samples, with an event rate of only 0.9%. Despite this extreme class imbalance, the model achieved an AUROC of 0.843 and an AUPRC of 0.535, with an overall accuracy of 91.1%. At an operating threshold of 0.011, the model yielded an F1-score of 0.113, corresponding to 1,446 TN, 137 FP, 5 FN, and 9 TP.

For adversarial optimization, we limited the total number of attack iterations per run to $T = 500$ and used $B = 30$ Monte Carlo samples for gradient estimation. Detailed hyperparameter settings for the adversarial optimization are provided in Table 13.

*Table 13.* Hyperparameters for VitalDB Attack. In this experiment, $\eta_0$, $\sigma$, and $\theta$ denote the initial learning rate, the initial smoothing scale, and the amplification parameter, respectively. Both ProMoT and ProMoT-loo used the same Gaussian smoothing kernel and shared an identical transformation function defined as $g(\theta, y) = (y + c)^\beta e^{\theta y}$, where $c = 1,000$ and $\beta = 10$. Following the baseline configurations, $\gamma$ represents the decreasing factor for the smoothing scale $\sigma$ in RSGF and ZO-SLGHd/r, and $\alpha$ denotes the step size used for updating the smoothing scale in ZO-SLGHd.

| Method | Hyperparameters |
|---|---|
| ProMoT | $\eta_0 = 0.01$, $\sigma = 0.3$, $\theta = 5$ |
| ProMoT-loo | $\eta_0 = 0.01$, $\sigma = 0.3$, $\theta = 5$ |
| EPGS | $\eta_0 = 0.03$, $\sigma = 0.3$, $\theta = 5$ |
| RSGF | $\eta_0 = 0.03$, $\sigma = 3.0$, $\gamma = 0.95$ |
| ZO-SGD | $\eta_0 = 0.3$, $\sigma = 0.3$ |
| ZO-AdaMM | $\eta_0 = 0.5$, $\sigma = 0.3$ ,$\beta_1 = 0.7$, $\beta_2 = 0.3$ |
| ZO-SLGHd | $\eta_0 = 0.1$, $\sigma = 1.0$, $\gamma = 0.999$, $\alpha = 0.001$ |
| ZO-SLGHr | $\eta_0 = 0.1$, $\sigma = 0.3$, $\gamma = 0.995$ |
| CMA-ES | $\sigma = 0.5$ |

