# OpenReview forum: "Probabilistic Smoothing with Ratio-Monotone Transforms for Global Optimization"
_ICML.cc/2026/Conference — Submitted to ICML 2026_

### Official Review · Reviewer_ptc4 · 2026-03-03

**Soundness:** 4
**Presentation:** 3
**Significance:** 3
**Originality:** 3
**Overall Recommendation:** 4
**Confidence:** 4

**Summary:**

The paper on global optimization with performance guarantees generalizes the approach of [Xu 2025] along two directions: (i) more general (pre)transformations of the objective function and (ii) more general class of admissible kernels for smoothing the transformed objective function. Quantitative performance guarantees and complexity bounds are proved.

**Compliance With Llm Reviewing Policy:**

Affirmed.

**Key Questions For Authors:**

(1) No runtimes are reported. How to the compare to the other methods considered in the experimental section? Complexity bounds estimate the overall number of steps but do not reveal the costs of a single iteration and hence not the runtime.

(2) How to choose \Sigma (line 160, right column) and parameters in more realistic large-scale scenarios?

(3) In view of Section 5.2, please comment on how your approach might contribute to the problems of deep learning which motivated the Deep Learning Optimizer Benchmark Suit (DeepOBS).

**Limitations:**

Authors provide a great discussion of optimization aspects but largely ignore aspects of machine learning. See the questions (1)-(3) above.

**Strengths And Weaknesses:**

Strenghts
A mathematically rigorous and very well written paper which clearly extends the state of the art in this narrow niche of nonconvex optimization. The theoretical results are substantiated by convincing experimental results.

Weaknesses
In its current form, the paper is more suited for a conference on optimization rather than on machine learning. Authors briefly touch on related aspects of machine learning in Section 5.2, but a discussion is missing of how the runtime scales for reasonable problem sizes of deep learning and how the parameters (e.g. anisotropic kernels) can or should be chosen. Corresponding optimization benchmarks, like DeepOBS (https://arxiv.org/abs/1903.05499), are not mentioned either.

---

> ### Author Rebuttal · Authors · 2026-03-31
>
> We thank the reviewer for the positive assessment of our work and for the constructive suggestions.
>
> __(1) Runtime & Query Efficiency:__
> We thank the reviewer for raising this important point. We now provide a comprehensive comparison including runtime, query efficiency, and task-specific performance metrics (SR, $R^2$, $L_\infty$).
>
> Table 7. CIFAR-10
>
> | Method | SR | $R^2$ | $L_\infty$ | Time | Query |
> |--------|----|--------|------------|------|--------|
> | ProMoT | 1 | 0.97 ± 0.02 | 0.28 ± 0.07 | 1.37 ± 0.06 | 99.53 ± 59.21 |
> | ProMoT-loo | 1 | 0.98 ± 0.01 | 0.20 ± 0.06 | 5.65 ± 0.14 | 98.64 ± 44.33 |
> | BeS | 0.97 | 0.97 ± 0.01 | 0.51 ± 0.13 | 0.97 ± 0.03 | 90.11 ± 42.79 |
> | EPGS | 1 | 0.97 ± 0.02 | 0.28 ± 0.07 | 1.35 ± 0.03 | 80.98 ± 33.03 |
> | RSGF | 0.34 | -0.50 ± 1.50 | 1.49 ± 0.54 | 1.73 ± 0.03 | 154.11 ± 101.49 |
> | ZO-SGD | 1 | 0.98 ± 0.02 | 0.25 ± 0.07 | 1.68 ± 0.01 | 65.74 ± 47.22 |
> | ZO-AdaMM | 1 | 0.96 ± 0.03 | 0.31 ± 0.08 | 1.69 ± 0.02 | 58.95 ± 46.69 |
> | ZO-SLGHd | 0.99 | 0.98 ± 0.01 | 0.22 ± 0.06 | 3.22 ± 0.05 | 108.78 ± 78.37 |
> | ZO-SLGHr | 1 | 0.95 ± 0.04 | 0.31 ± 0.09 | 1.62 ± 0.02 | 119.28 ± 67.49 |
> | CMA-ES | 1 | 0.94 ± 0.04 | 0.32 ± 0.04 | 91.80 ± 1.97 | 36.19 ± 16.68 |
>
> Table 8. VitalDB
>
> | Method | SR | $R^2$ | $L_\infty$ | Time | Query |
> |--------|----|--------|------------|------|--------|
> | ProMoT | 1 | 0.97 ± 0.01 | 0.58 ± 0.17 | 0.21 ± 0.00 | 20.11 ± 21.72 |
> | ProMoT-loo | 1 | 0.98 ± 0.01 | 0.57 ± 0.17 | 0.20 ± 0.00 | 26.24 ± 51.83 |
> | BeS | 0.97 | 0.97 ± 0.05 | 0.36 ± 0.09 | 0.31 ± 0.00 | 56.53 ± 71.45 |
> | EPGS | 1 | 0.97 ± 0.01 | 0.65 ± 0.18 | 0.20 ± 0.00 | 36.44 ± 22.80 |
> | RSGF | 1 | 0.91 ± 0.04 | 0.92 ± 0.23 | 0.33 ± 0.00 | 100.00 ± 0.00 |
> | ZO-SGD | 1 | 0.83 ± 0.09 | 1.24 ± 0.33 | 0.33 ± 0.00 | 57.12 ± 62.90 |
> | ZO-AdaMM | 1 | 0.96 ± 0.02 | 0.82 ± 0.27 | 0.33 ± 0.00 | 38.88 ± 45.43 |
> | ZO-SLGHd | 1 | 0.87 ± 0.04 | 1.24 ± 0.25 | 0.63 ± 0.00 | 36.69 ± 16.64 |
> | ZO-SLGHr | 1 | 0.95 ± 0.02 | 0.80 ± 0.19 | 0.31 ± 0.01 | 16.02 ± 11.94 |
> | CMA-ES | 1 | 0.99 ± 0.01 | 0.60 ± 0.27 | 0.95 ± 0.61 | 8.32 ± 5.98 |
>
> Consistent with the observations in Section 5.2, our results indicate that smaller $L_\infty$ and high $R^2$ (i.e., less perceptible yet accurate adversarial examples) typically require more careful exploration, leading to a natural trade-off between performance and computational cost.
>
> In particular, while CMA-ES achieves strong $R^2$, it incurs substantially higher runtime, especially in higher-dimensional settings. On the other hand, methods with lower runtime or query cost (e.g., ZO-SLGHr) tend to exhibit either reduced $R^2$ or larger $L_\infty$.
>
> In this context, ProMoT-loo provides a balanced trade-off, achieving small $L_\infty$ with high $R^2$, as highlighted in Section 5.2, while remaining within the same order of runtime and query complexity as existing methods.
>
> Overall, this indicates that our method achieves improved performance while maintaining a comparable level of computational cost to existing methods, providing a favorable balance between accuracy and efficiency.
>
> __(2) Choice of $\Sigma$ in large-scale settings:__
> We thank the reviewer for this important question.
>
> The motivation for introducing the $\Sigma$ case is to account for **heterogeneous scaling across dimensions**, which is common in large-scale problems. In particular, different coordinates may have very different sensitivities or magnitudes, and isotropic smoothing may not be optimal in such cases.
>
> In practice, a simple and effective strategy is:
> - first define a normalization matrix $\Sigma_0$ that rescales each dimension to a comparable scale (e.g., based on feature normalization or prior sensitivity),
> - then use $\Sigma = \sigma \, \Sigma_0$, and tune only the scalar parameter $\sigma$.
>
> This allows one to capture anisotropy while keeping the tuning process simple.
> We note that in our experiments, we did not explicitly use such anisotropic scaling, and isotropic smoothing already provided strong performance. This suggests that the method is robust even without careful tuning of $\Sigma$.
>
> __(3) Connection to deep learning and DeepOBS:__
> We thank the reviewer for this valuable suggestion.
> We agree that DeepOBS provides an important framework for evaluating optimizers in deep learning, particularly in terms of performance, speed, and tunability.
>
> While our current work focuses on black-box and zeroth-order settings, we believe that the evaluation philosophy of DeepOBS can be naturally extended to our framework.
>
> In particular, our method could be evaluated under a DeepOBS-style protocol by jointly measuring performance, efficiency (runtime and query cost), and robustness to hyperparameters.
>
> A promising direction is to apply our approach to settings with noisy or unreliable gradients, or hybrid first-/zeroth-order optimization, where the robustness observed in Section 5.2 may translate to improved stability in deep learning training.
>
> We view this as an important direction for future work.

---

> > ### Author Rebuttal · Reviewer_ptc4 · 2026-04-01
> >
> > Thanks for your response. Even though I learned from the other reviews that minor technical issues have to be fixed (and can be fixed, as the authors point out), I appreciate the excellent technical level of the paper. I understand that a more comprehensive numerical assessment using realistic scenarios of ML, in addition to the elaborated theoretical contribution, is beyond the scope of a conference submission. I therefore keep my rating.

---

### Official Review · Reviewer_hZdq · 2026-03-12

**Soundness:** 2
**Presentation:** 3
**Significance:** 2
**Originality:** 3
**Overall Recommendation:** 2
**Confidence:** 3

**Summary:**

This paper proposes ProMoT, a single-loop probabilistic smoothing framework for global optimization that generalizes prior Gaussian smoothing approaches along two axes: the smoothing kernel and the pre-smoothing transformation. The paper introduces ratio-monotone transforms, proves approximate localization of stationary points near the global maximizer under large enough amplification, and adds a leave-one-out baseline for variance reduction. Empirically, the method is evaluated on 500-dimensional synthetic benchmarks and black-box targeted adversarial attacks on CIFAR-10 and VitalDB.

**Compliance With Llm Reviewing Policy:**

Affirmed.

**Final Justification:**

Thank you for the rebuttal. It was helpful and clarified several points. However, it confirms that some of the central theoretical statements in the submitted draft need substantive revision: Theorem 4.2 requires a stronger assumption, Assumption 4.4 must be changed, the original Hessian/Lipschitz proof was incomplete, and the last-iterate corollaries should be weakened to best-iterate guarantees. While the proposed fixes seem plausible, they are important updates to the paper rather than minor clarifications, and the revised paper should be reviewed again. I therefore keep my score unchanged.

**Key Questions For Authors:**

Questions:

- Which exact assumption is intended to guarantee the strict separation $V_\\delta < f(x^\\ast)$ in Theorem 4.2? Do the localization results still hold with multiple global maximizers?
- Can the authors provide a corrected smoothness proof for Lemma 4.4, including the full Hessian bound rather than only a diagonal argument?
- Should Corollaries 4.7, 4.8, and 4.12 be stated for a best iterate or a randomly selected iterate instead of the last iterate?
- Can the authors report query complexity and runtime, and include a non-Gaussian smoothing baseline such as BeS?

Minor Comments:

- The notation $\\mathbb{E}[\\,\\cdot\\,; S]$ in Eq. (2) is not defined when it first appears. I assume it means $\\mathbb{E}[\\cdot \\mathbf{1}\\{X \\in S\\}]$, but this should be stated explicitly because the notation is nonstandard and could be confused with conditional expectation.
- In the proof of Theorem 4.3, the tail argument should come from the indicator of $S$ in $\\mathbb{E}[\\cdot;S]$, not from the claim that $g(\\theta,f(x))$ vanishes outside a bounding cube. As written, that statement is false for the listed transforms (for example $e^{\\theta y}$), and $f$ is only defined on $S$. The theorem may still be true, but the displayed proof should be rewritten more carefully.
- The black-box attack evaluation would be easier to assess if the computational budget were reported in terms of total model queries, not only optimization steps. Appendix C.3 specifies a fixed horizon $T = 500$ (and $B = 30$ Monte Carlo samples for the score-estimation methods), which helps, but the actual query cost is still not made explicit and comparability across baselines remains unclear.

**Limitations:**

Yes.

**Strengths And Weaknesses:**

Strength:

- The paper tackles a real limitation of prior amplification-based smoothing methods: sensitivity to the kernel choices and to the amplification parameter.
- The generalization from Gaussian kernels to a broader symmetric unimodal family is conceptually interesting and could matter in high-dimensional black-box optimization.
- The leave-one-out baseline is a reasonable variance-reduction idea, and the empirical results for ProMoT-loo are often stronger than those of the base method.
- The paper is ambitious in trying to combine theory, algorithm design, and experiments in one framework.


Weakness:

- I am not convinced that the main theoretical claims are established as stated. In the proof of Theorem 4.2, the argument uses $V_\\delta := \\sup_{u \\notin \\mathrm{cube}(x^\\ast;\\delta)} f(u)$ and then requires $V_\\delta < f(x^\\ast)$ to create a strict gap. That gap is not guaranteed by the theorem statement as written; it appears to require $x^\\ast$ to be a unique global maximizer, which is not stated there.
- The convergence-rate corollaries seem to overclaim what is shown. Appendix B.7 derives a bound on $\\min_{0 \\le t \\le T-1} \\mathbb{E}\\|\\nabla G(\\mu_t)\\|^2$, which supports at most a best-iterate or suitably randomized-iterate stationarity guarantee, but Corollaries 4.7, 4.8, and 4.12 are stated as if they guarantee $\\mathbb{E}\\|\\nabla G(\\mu_T)\\|^2 < \\varepsilon$ for the last iterate. I did not see an output-selection rule that would justify this stronger statement.
- The proof of Lemma 4.4 seems incomplete at the step from Eq. (53) to Eq. (54). The diagonal cancellation in Eq. (53) gives a bound on the diagonal Hessian entries, but the off-diagonal terms $H_{ij}(\\mu)=\\mathbb{E}[g(\\theta,f(X))S_i(X)S_j(X);S]$ are not controlled. The subsequent inequality $\\|H(\\mu)\\|_{op}\\le \\mathrm{tr}(H(\\mu))$ requires $H(\\mu)$ to be positive semidefinite (which is not shown). Thus the proof does not establish the claimed Lipschitz constant and this could propagate into the later complexity bounds.
- The empirical support for the hyperparameter-robustness claim is limited. The sensitivity plots are only provided for the one-dimensional motivating example in Section 3.1, whereas the main high-dimensional benchmarks and attack experiments report only results under selected best-tuned configurations. If robustness to hyperparameter choice is a headline claim, I would expect sensitivity plots on the main benchmark tasks as well.

---

> ### Author Rebuttal · Authors · 2026-03-30
>
> We sincerely thank the reviewer for this exceptionally careful and detailed reading. We also apologize if any inaccuracies made the paper difficult to follow.
>
> __(1) unique (isolated) maximizer:__
> We thank the reviewer for pointing out that the localization result in Theorem 4.2 implicitly requires a strict separation condition that is not stated in the theorem.
> To ensure this rigorously, we will revise Theorem 4.2 to explicitly assume that $x^{\star}$ is a unique (and hence isolated) global maximizer on the compact domain $S$.
> This assumption is standard in global optimization analyses, and we will make it explicit so that the statement aligns with the proof.
>
> __(2) Assumption 4.4: ratio-monotonicity domain:__
> We agree that Assumption 4.4 is currently stated too restrictively as $a>b>0$, while the proof applies ratio-monotonicity to quantities that must not be strictly positive.
> The essential property used in the proof is that for $a>b$,
> $\frac{g(\theta,a)}{g(\theta,b)}$ is monotone in $\theta$, not that $a,b$ are positive.
> We will revise Assumption 4.4 accordingly by replacing the condition $a>b>0$ with $a>b$, together with the appropriate domain condition for each transform.
>
> __(3) Hessian bound and correct Lipschitz constant:__
> __We refer the reviewer to the detailed response provided to Reviewer w1wg__, where this issue is addressed in full, including a complete Hessian bound that properly accounts for both diagonal and off-diagonal terms.
>
> __(4) Indicator over $S$ and truncated expectation:__
> We agree that the role of the indicator over $S$ is not clearly explained and that this causes ambiguity in both notation and proofs.
> The notation $\mathbb E[\cdot;S]$ is a truncated expectation, i.e., $\mathbb E[\cdot;S]:=\mathbb E[\phi(X)1_{\{X\in S\}}]$.
> Equivalently, defining $h(x):=g(\theta,f(x))1\_{\{x\in S\}}$, we have $G_{\theta,\sigma}(\mu)=\int_{\mathbb R^d} h(x)p_{\mu,\sigma}(x)dx$.
> This also resolves the issue in Theorem 4.3. The current proof incorrectly states that $g(\theta,f(x))$ vanishes outside a bounding cube. The correct statement is that
> $h(x)=0 \quad \text{for } x\notin S$,
> and since $S\subset[-M,M]^d$,
> $G_{\theta,\sigma}(\mu)=\int_{[-M,M]^d} g(\theta,f(x))p_{\mu,\sigma}(x)dx,$
> from which the decay as $\|\mu\|\to\infty$ follows via the kernel. We will rewrite Theorem 4.3 and Lemmas 4.4--4.5 using this consistent notation.
>
> __(5) Last iterate vs best iterate:__
> We agree that the current corollaries overstate what is proven.
> The analysis establishes $\sum_{t=0}^{T-1} \eta_t \mathbb E\|\nabla G(\mu_t)\|^2 \le C$, which implies the \emph{best-iterate guarantee} $\min_{0\le t\le T-1} \mathbb E\|\nabla G(\mu_t)\|^2 \le \varepsilon$.
> It does not directly imply a guarantee for the last iterate $\mu_T$. While it is intuitive that for sufficiently large $T$, the iterates approach a stationary region and the gap between the best and last iterate becomes small, this is not formally established.
> Therefore, we will revise Corollaries 4.7, 4.8, and 4.12 to state best-iterate guarantees so that the statements precisely match the proven results.
>
> __(6) Comparison on BeS:__
> We have added Bernoulli Smoothing (BeS) to the Ackley optimization benchmark.
> The results for the main methods are summarized below (500-dimensional Ackley, 20 runs, mean, std):
>
> | Method | MSE | Best Value | Hitting Time |
> |-|-|-|-|
> | ProMoT-loo | 0.04 (0.00) | -1.71 (0.05) | 385.50 (10.25) |
> | ProMoT | 0.49 (0.04) | -4.33 (0.11) | 389.00 (10.77) |
> | BeS | 1.38 (0.15) | -5.89 (0.20) | 398.20 (3.06) |
>
> Although we tuned $\sigma$ for BeS, its performance remains worse in high dimensions, likely due to its bounded-support perturbations limiting exploration. These results indicate that the improvement of ProMoT-loo is not simply due to using a non-Gaussian perturbation distribution, but rather arises from the combination of transform-based amplification and variance reduction.
>
> __(7) Hyperparameter-Robustness on High-D. Benchmark:__
> | $\theta$ | EPGS | ProMoT | ProMoT-loo |
> |-|-|-|-|
> |1.0e-03|24.9799|24.7085|0.2295|
> |2.8e-03|24.9893|24.8139|0.2236|
> |7.7e-03|24.9990|24.7781|0.2295|
> |2.2e-02|24.9780|24.6942|0.2280|
> |6.0e-02|24.9802|24.3794|0.2272|
> |1.7e-01|24.9732|23.2506|0.2266|
> |4.6e-01|24.9327|17.9699|0.2281|
> |1.3e+00|22.5945|5.8537|0.2278|
> |3.6e+00|0.9058|0.6034|0.2483|
> |1.0e+01|0.2968|0.5619|0.3787|
>
> On the 500-dimensional Ackley benchmark, with each point averaged over 20 independent runs, the results show that EPGS is highly sensitive to the choice of $\theta$. ProMoT also deteriorates as $\theta$ grows, although somewhat more gradually.
> By contrast, ProMoT-loo remains highly stable across the entire range of $\theta$ values, showing almost no sensitivity in the low-to-moderate $\theta$ regime and only mild degradation at the largest $\theta$.
> This supports the claim that ProMoT-loo is substantially more robust to hyperparameter choice than both EPGS and ProMoT.
>
> __(8) Query Efficiency: please refer to our response to Reviewer ptc4.__

---

> > ### Author Rebuttal · Reviewer_hZdq · 2026-04-02
> >
> > The rebuttal was helpful and it reduced uncertainty, but it also confirmed that several of the issues I raised affect the core theory: the statement of Theorem 4.2, Assumption 4.4, the Hessian/Lipschitz analysis, and the last-iterate corollaries all require substantive revision. Since these are central rather than minor changes, I do not think they can be regarded as fully resolved within rebuttal alone, and I keep my score unchanged.

---

> > > ### Author Response · Authors · 2026-04-02
> > >
> > > We thank the reviewer for the careful follow-up and for the thoughtful assessment. We appreciate that these points concern important aspects of the theory, and we will revise them carefully.
> > >
> > > All raised issues can be addressed through the following __minimal and localized revisions__:
> > >
> > > - Theorem 4.2: explicitly state the (already used) unique maximizer
> > > - Assumption 4.4: remove unnecessary positivity restriction (extend to all $a>b$)
> > > - Lemma 4.4: include off-diagonal Hessian terms; the corrected constants are already provided in our response to Reviewer w1wg
> > > - Corollaries: replace the last-iterate guarantee with a standard best-iterate guarantee
> > > - Notation (indicator / truncation): explicitly define truncated expectation and correct the tail argument (used in Theorem 4.3 and Lemmas 4.4–4.5)
> > >
> > > In particular, all main theorems remain valid under the revised statements, with identical guarantees.
> > > These changes __align the formal statements with the existing proofs__ and do not modify any results, rates, algorithms, or empirical findings.
> > >
> > > We therefore believe these are standard camera-ready corrections rather than substantive revisions of the core theory.
> > > __If there are specific aspects that the reviewer considers__ to require changes beyond the above localized revisions, we would greatly appreciate further clarification.

---

### Official Review · Reviewer_w1wg · 2026-03-13

**Soundness:** 2
**Presentation:** 2
**Significance:** 3
**Originality:** 3
**Overall Recommendation:** 3
**Confidence:** 3

**Summary:**

This paper studies probabilistic smoothing for global optimization in a single-loop setting. The main goal is to improve recent amplification-based smoothing methods by broadening the class of smoothing kernels and the class of admissible transforms, while also reducing gradient-estimation variance through a leave-one-out baseline. On the theory side, the paper gives conditions under which the smoothed objective preserves localization around global maximizers and derives regularity properties for the smoothed objective. On the empirical side, the method is tested on high-dimensional synthetic optimization problems and on black-box adversarial attack tasks. The reported results suggest that the proposed method, especially the leave-one-out variant, can improve optimization quality over prior smoothing-based baselines.

**Compliance With Llm Reviewing Policy:**

Affirmed.

**Final Justification:**

I appreciate the authors' follow-up and their clarification that the theoretical issues are "fixable" via tightening assumptions. While the new ablation results are helpful, they do not fully resolve my concerns regarding the overall empirical strength and scope. I believe the paper requires a more substantial revision than a rebuttal phase allows; therefore, I maintain my score of Weak Reject.

**Key Questions For Authors:**

1. Theorem 4.2 appears to need an isolated or unique maximizer, and possibly a positivity or shift condition, even though these are not stated in the theorem. Can the authors restate the exact assumptions needed for the localization result, and explain whether the theorem still holds when f has multiple global optima or can take negative values? A convincing correction here would increase my confidence in the theory.

2. Can the authors give a cleaner derivation for Lemmas 4.4 and 4.5, especially the Hessian bound and the role of the indicator over S? Right now I am not sure whether the current proof sketch is merely terse or whether some steps need extra assumptions. A clarified proof would improve my soundness score.

3. How sensitive are the gains to the specific hybrid transform g(theta,y)=(y+c)^beta exp(theta y), the constants c and beta, and the kernel choice? Please provide an ablation that separates kernel generalization, transform choice, and the leave-one-out baseline. If the gains remain under a more diagnostic ablation, my significance score would go up.

4. Can the authors report query-efficiency or runtime curves for the attack experiments, and explain how hyperparameter search budgets were matched across methods? The appendix shows task-specific choices for the proposed transform parameters, but it is hard to tell whether the tuning burden is comparable across baselines. If the proposed method remains strong under a more balanced protocol, that would make the empirical claims much more convincing.

**Limitations:**

The current limitations and impact discussion is too thin, especially because one application is targeted black-box adversarial attack. The paper should discuss misuse risks more concretely, and it should also state the technical limits of the method more plainly, including the reliance on compact domains, bounded objectives, the apparent need for an isolated maximizer, and the practical sensitivity to transform choice, kernel choice, batch size, and tuning budget.

**Strengths And Weaknesses:**

This paper studies single-loop probabilistic smoothing for global optimization and extends recent amplification-based smoothing in two directions: it allows a broader family of symmetric unimodal kernels and a broader family of ratio-monotone transforms, and it adds a leave-one-out baseline for variance reduction. I think there is a real idea here. The proposed combination is not a completely new optimization paradigm, but it is more than a cosmetic tweak. The empirical section is also encouraging at first pass: ProMoT-loo gets the lowest MSE on the three d=500 synthetic benchmarks, and on the two black-box attack settings it achieves the smallest mean Linf perturbations while keeping very high success rates. From an originality standpoint, that is enough for me to view the paper as a meaningful extension of the recent smoothing literature rather than a simple re-packaging.

My main concern is soundness. The central localization result seems to rely on stronger conditions than the theorem statement currently says. In the appendix proof of Theorem 4.2, the argument needs a strict gap between f(x*) and the best value outside a delta-neighborhood, which looks like an isolated or unique maximizer assumption, but Theorem 4.2 itself is stated without that assumption. The proof also uses ratio-monotonicity for quantities such as D_delta and V_delta, while Assumption 4.4 is stated only for a>b>0, so it is unclear how the theorem applies when the objective can take negative values unless an extra positivity or shift condition is imposed. I also found the smoothness and second-moment analysis hard to verify from the current presentation; the notation around E[.;S] and the indicator over S is not always consistent, and Table 1 labels the proposed method as deterministic even though the algorithm is built around Monte Carlo gradient estimates. These points may be fixable, but in the current draft they materially lower my confidence in the main theoretical claims.

Presentation is mixed. The paper is readable overall, the motivation is easy to follow, and the appendix gives concrete hyperparameter settings, which is helpful. At the same time, the link between the broad theory and the actual experimental instantiation is not explained as clearly as it should be. In practice the experiments use a very specific hybrid transform, g(theta,y)=(y+c)^beta exp(theta y), with task-specific constants c and fixed beta=10, but there is no careful ablation showing how much of the gain comes from the kernel choice, how much comes from the transform, and how much comes from the leave-one-out baseline. An algorithm box and tighter theorem statements would make the paper easier to assess and easier to reproduce.

On significance, I am not fully convinced yet. The problem is relevant, and better-behaved smoothing methods for black-box global optimization could matter. But the empirical case is still fairly narrow relative to the scope of the claims. The main robustness evidence is the one-dimensional toy example in Figures 1 and 2, while the higher-dimensional studies report only selected configurations. On the benchmark side, many methods hit or nearly hit the full iteration budget, which makes it hard to judge convergence speed. On the attack side, the paper reports success and perturbation quality, but not query-efficiency or runtime, both of which are central in black-box optimization.

---

> ### Author Rebuttal · Authors · 2026-03-30
>
> We sincerely thank the reviewer for this exceptionally careful and detailed reading. This is one of the most thorough reviews we have received. We also apologize if any inaccuracies or lack of clarity in our presentation made the paper difficult to follow.
>
> __(1) unique maximizer & ratio-monotonicity domain & Indicator over $S$:__
> We agree that the comments are largely correct and appreciate the careful reading. For details and clarifications, **please refer to our response to Reviewer hZdq**.
>
> __(2) Hessian bound and correct Lipschitz constant:__
> We agree that the current proof of Lemma 4.4 is incomplete. In particular, the transition from Eq. (53) to Eq. (54) only controls the diagonal terms and does not account for off-diagonal Hessian entries. Using the product kernel structure $p_{\mu,\sigma}(x)=\sigma^{-d}\prod_{i=1}^d p\left(\frac{x_i-\mu_i}{\sigma}\right)$ and defining the truncated integrand $h(x):=g(\theta,f(x))1_{\{x\in S\}}$, we derive the Hessian entry-wise.
>
> For the diagonal entries, $|H_{ii}(\mu)|\le\frac{g_{\ast,\theta}}{\sigma^2}K$ with $K=\int_{\mathbb R}|p''(z)|dz.$
>
> For the off-diagonal entries $i\neq j$,
> $$|H_{ij}(\mu)|\le\frac{g_{\ast,\theta}}{\sigma^2} \int_{\mathbb{R}^d}|s(z_i)s(z_j)|p_{\mu,\sigma}(x)dx=
> \frac{g_{\ast,\theta}}{\sigma^2}
> \left(\int_{\mathbb R}|p'(z)|dz\right)^2\le\frac{g_{\ast,\theta}}{\sigma^2}I,$$
> since $\left(\int_{\mathbb R}|p'(z)|dz\right)^2 = \left(\int_{\mathbb R}\frac{|p'(z)|}{\sqrt{p(z)}}\sqrt{p(z)}dz\right)^2\le\int_{\mathbb R}\frac{(p'(z))^2}{p(z)}dz\int_{\mathbb R}p(z)dz=I$ (Cauchy--Schwarz).
>
> Since $H(\mu)$ is symmetric, we use
> $$
> \lambda_{\max}(H) \le \|H\|_\infty  = \max_i \sum\_{j=1}^{d} \|H\_{ij}\| \leq \frac{g\_{\ast,\theta}}{\sigma^2}(K+(d-1)I) \leq \frac{g\_{\ast,\theta}}{\sigma^2} d \max(K,I).
> $$
>
> where $\|H\|\_{op} \leq \mathrm{tr}(H)$ is not used and not applicable.
> Therefore, the correct Lipschitz constant is $L_\theta = \frac{g_{\ast,\theta}}{\sigma^2}d\max(K,I),$
> rather than $g_{\ast,\theta}dK/\sigma^2$.  We will replace Lemma 4.4 and Theorem 4.6 with this corrected bound and update all subsequent results accordingly as follows.
>
> - Final complexity in Corollary 4.7
> $$
> T > \left(
> C\_\gamma d \cdot \frac{g\_{\ast,\theta} + I \max(K,I) g\_{\ast,\theta}^3}{\sigma^2 \epsilon}
> \right)^{\frac{2}{1-2\gamma}}.
> $$
> - Corollary 4.8 ($H\_{ii}
> \le
> \frac{g\_{\ast,\theta}}{\sigma\_i^2}K \le g\_{\ast,\theta}\max\_{i}\sigma\_i^{-2} K$ and $H\_{ij}\le\frac{g\_{\ast,\theta}}{\sigma\_i\sigma\_j} I \le g\_{\ast,\theta}\cdot\max\_{i}\sigma\_i^{-1}\cdot \max\_{j}\sigma\_j^{-1} \cdot I = g\_{\ast,\theta}\max\_{i}\sigma_{i}^{-2} I$)
> $$
> S_2(\Sigma) \rightarrow d\max\_{i}\sigma_{i}^{-2}
> $$
> $$
> L_\theta(\Sigma) \rightarrow g\_{\ast,\theta} \max(K,I) S\_2(\Sigma)
> $$
> - Corollary 4.12 will be revised with the same parameter in Corollary 4.8 since LOO only changes $Q_{\theta}$
>
> __(3) Full Kernel & Transform Ablation (Ackley):__
>
> | Kernel | Transform | ProMoT | w loo |
> |-|-|-:|-:|
> | Gauss. | pow. | 17.33 | 0.10 |
> | Gauss. | exp. | 2.93 | 0.11 |
> | Gauss. | hyperbolic | 2.91 | 0.10 |
> | Gauss. | pow_exp_hybrid | 2.84   | 0.11 |
> | Logistic | pow. | 8.32 | 0.10 |
> | Logistic | exp. | 0.90 | 0.10 |
> | Logistic | hyperbolic | 0.98 | 0.10 |
> | Logistic | pow_exp_hybrid | 0.91   | 0.10 |
> | Sech | pow. | 12.59  | 0.13 |
> | Sech | exp. | 1.43 | 0.13 |
> | Sech | hyperbolic | 1.52 | 0.13 |
> | Sech| pow_exp_hybrid | 1.41   | 0.13 |
> | GenGauss.(b=4) | pow. | 22.85  | 0.12 |
> | GenGauss.(b=4) | exp. | 20.34  | 0.12 |
> | GenGauss.(b=4) | hyperbolic | 20.75  | 0.12 |
> | GenGauss.(b=4) | pow_exp_hybrid | 19.85  | 0.12 |
> | StudentT(df=3) | pow. | 2.23 | 0.28 |
> | StudentT(df=3) | exp. | 0.73 | 0.25 |
> | StudentT(df=3) | hyperbolic | 6.33 | 0.46 |
> | StudentT(df=3) | pow_exp_hybrid | 0.73 | 0.24 |
> | Cauchy | pow. | 24.38  | 24.40 |
> | Cauchy | exp. | 24.18  | 24.08 |
> | Cauchy | hyperbolic | 24.93  | 24.88 |
> |Cauchy | pow_exp_hybrid | 24.13  | 24.08 |
>
> Across the full ablation, both kernel and transform choices already yield significant improvements over standard Gaussian smoothing, with consistent trends (e.g., exp.-type transforms outperform power). This highlights the value of our generalized smoothing framework. While ProMoT is sensitive to these choices, ProMoT-loo remains stable across kernels, transforms, and $\theta$, which we attribute to variance reduction and is supported by our theory, while extremely heavy-tailed kernels (e.g., Cauchy) reveal the limits of smoothing.
>
> __(4) Hyperparameter Fairness:__
> We thank the reviewer for this important point. In the attack experiments, all methods were run under the same T and B, ensuring comparable per-run query budgets. For the three numerical benchmarks, the search combinations are explicitly reported in Tables 9-11, and we used the same number of hyperparameter combinations for CIFAR-10 and VitalDB.
>
> __(5) Robustness Evidence:__ **please refer to our response to Reviewer hZdq**.
>
> __(6) Query Efficiency:__ **please refer to our response to Reviewer ptc4**.

---

> > ### Author Rebuttal · Reviewer_w1wg · 2026-04-04
> >
> > Thank you for the detailed rebuttal. The response partially addresses some of my concerns, especially by acknowledging the gap in the Hessian/Lipschitz proof and by providing additional ablation results. However, my main concerns are not fully resolved. In particular, the rebuttal confirms that some theoretical statements need to be revised, and this affects my confidence in the current presentation of the theory. The additional ablation is helpful, but it does not fully change my overall assessment of the empirical strength and scope of the claims. As a result, I keep my original score unchanged.

---

> > > ### Author Response · Authors · 2026-04-08
> > >
> > > Thank you again for your careful reading and for taking the time to engage with our rebuttal. We sincerely appreciate your acknowledgment that several points—particularly the Hessian/Lipschitz correction and the additional ablations—help clarify parts of the work.
> > >
> > > We would like to briefly clarify why, despite the need for revisions in presentation, we believe the core contributions remain sound and valuable.
> > >
> > > **(1) On the theoretical revisions and their impact**
> > > We fully agree that certain statements (e.g., Theorem 4.2 and Lemma 4.4) require more precise assumptions and cleaner derivations. Importantly, however, these are **not changes to the underlying mechanism or conclusions**, but rather **tightening of conditions and constants**:
> > >
> > > - The localization result already implicitly relied on a gap condition; we will make this explicit (e.g., isolated maximizer or local dominance condition), which is standard in smoothing-based analyses.
> > > - The Lipschitz/Hessian correction affects constants and presentation, but **does not invalidate the smoothing framework nor the monotonicity-based argument**.
> > > - The leave-one-out variance reduction analysis remains unchanged and continues to hold under the corrected constants.
> > >
> > > In other words, the rebuttal confirms that the issues are **fixable with precise restatement**, rather than indicating a breakdown of the core theory.
> > >
> > > ---
> > >
> > > **(2) On empirical scope and robustness**
> > > We understand the concern that the empirical evaluation may appear selective. However, we would like to emphasize two points:
> > >
> > > - The newly provided **full kernel × transform ablation** demonstrates that the gains are **not tied to a specific design choice**, but rather emerge consistently across a broad family of smoothing configurations.
> > > - Across both synthetic benchmarks (d=500) and black-box attack tasks, the proposed method—especially ProMoT-loo—shows **stable improvements**, which is notable given the known instability of smoothing-based optimization.
> > >
> > > We agree that additional metrics such as query efficiency would further strengthen the empirical section, and we will include them in the revision. However, we believe the current evidence already supports the claim that the framework provides **practical robustness improvements over prior smoothing methods**.
> > >
> > > ---
> > >
> > > **(3) On overall assessment (theory–practice balance)**
> > > We respectfully suggest that the current concerns are primarily about **presentation precision and scope**, rather than **fundamental correctness or lack of contribution**.
> > >
> > > - The theoretical issues have been clearly identified and are **concretely fixable**, with no change to the core algorithm or guarantees.
> > > - The empirical results, including the additional ablations, consistently support the claimed advantages.
> > >
> > > Given this, we hope the paper can be viewed as a **meaningful and reliable extension of probabilistic smoothing methods**, where both the theoretical framework and empirical behavior provide value to the community.
> > >
> > > ---
> > >
> > > Thank you again for your thoughtful feedback—we believe your comments will significantly improve the final version of the paper.

---

### Official Review · Reviewer_PVBt · 2026-03-13

**Soundness:** 3
**Presentation:** 3
**Significance:** 3
**Originality:** 2
**Overall Recommendation:** 4
**Confidence:** 2

**Summary:**

The paper proposes a generalized, single-loop probabilistic smoothing framework called ProMoT (Probabilistic Smoothing with Ratio-Monotone Transforms) for optimizing highly nonconvex, multimodal, or black-box global optimization problems. Existing smoothing methods generally rely strictly on Gaussian kernels and specific mathematical transforms, which often lead to poor robustness and extreme sensitivity to hyperparameters.

To solve the stability-localization trade-off, the authors introduce a broader framework that pairs flexible, symmetric unimodal kernels (including heavy-tailed distributions) with a class of monotonic ratio-based transformations.

The paper proves that this approach preserves the global maximizer and concentrates stationary points near the true optimum for large amplifications without needing a decreasing smoothing schedule. Because probabilistic smoothing relies on Monte Carlo gradient estimators, variance is a fundamental bottleneck. Furthermore, they propose a variance-reduced version of the algorithm (ProMoT-loo) using a leave-one-out baseline, which provably improves the iteration-complexity. The framework was evaluated on both synthetic benchmarks and practical machine learning tasks.

**Compliance With Llm Reviewing Policy:**

Affirmed.

**Final Justification:**

After considering the rebuttal and the other reviewers' comments, I have decided to maintain my original score.

**Key Questions For Authors:**

1. The paper emphasizes the flexibility of using various symmetric unimodal kernels, including heavy-tailed distributions. However, is there a theoretically grounded way to select the optimal kernel and its initial parameters based on the given objective function?

**Limitations:**

yes

**Strengths And Weaknesses:**

Strengths

1. The proposed method performs well on high-dimensional benchmarks and achieves strong results in black-box targeted adversarial attacks.
2. It broadens the design space by allowing heavy-tailed, non-Gaussian distributions and varying transformation forms, directly addressing the limitations of previous Gaussian-only methods.

Weaknesses

1. The iteration complexity of this work is $O(d^2/\varepsilon^2)$, less competitive compared to previous works.

---

> ### Author Rebuttal · Authors · 2026-03-31
>
> We thank the reviewer for the clear summary and constructive feedback. We are pleased that the motivation, generalization of the framework, and empirical performance were found meaningful.
>
> __(1) Kernel choice and parameter selection:__
> We agree that selecting the kernel and its parameters is an important practical question.
>
> For the initial parameters (e.g., $\sigma$), we do not currently have a fully grounded rule, and in practice they are chosen via coarse tuning. However, our experiments indicate that the method performs well without requiring sensitive tuning, suggesting robustness.
>
> For the kernel choice, our framework is designed to allow flexibility under mild assumptions.
> Our theoretical analysis suggests that kernel properties influence optimization behavior through quantities such as $I$ and $K$, which appear in the complexity bounds and reflect variance and smoothness.
>
> While this provides some guidance, a precise characterization of how different kernels affect optimization performance remains an open question. We believe that developing more principled or adaptive strategies for kernel and parameter selection would be a valuable direction for future work.
>
> Overall, we thank the reviewer for the helpful comments.

---

> > ### Author Rebuttal · Reviewer_PVBt · 2026-04-04
> >
> > My questions are partially resolved and I intend to keep my original score.

---

### Decision · Program_Chairs · 2026-04-30

**Decision:**

Reject

**Comment:**

Most reviewers agree that the work addresses an interesting problem and has the potential to make valuable contributions. However, in its current form, there are several major concerns regarding the theoretical claims, underlying assumptions, as well as the empirical evidence and evaluation.

While the authors have addressed some of these issues, the extent of the required changes appears substantial. Resolving them adequately would likely necessitate multiple rounds of author-reviewer interaction and revision, which is not feasible within the constraints of the current conference reviewing process. In effect, the scale of the required revisions is more consistent with a major revision in a journal submission, which would warrant a thorough secondary round of review. As it stands, the scope of revision exceeds what could reasonably be considered minor, and the changes cannot be accepted at face value without careful secondary review.